# Fluctuations of radio occultation signals in sounding the Earth's atmosphere

Valery Kan[1], Michael E. Gorbunov[1,2], and Viktoria F. Sofieva[3]

[1]A.M.Obukhov Institute of Atmospheric Physics Russian Academy of Sciences, 119017, Moscow, Pyzhevsky per. 3
[2]Hydrometeorological Research Centre of Russian Federation, 123242, Moscow, B. Predtechensky per., 11-13
[3]Finnish Meteorological Institute, Erik Palménin aukio 1, FI-00560, Helsinki, Finnland

*Correspondence to:* Michael Gorbunov (gorbunov@ifaran.ru)

**Abstract.** We discuss the relationships that link the observed fluctuation spectra of the amplitude and phase of signals used for the radio occultation sounding of the Earth's atmosphere, with the spectra of atmospheric inhomogeneities. Our analysis employs the approximation of the phase screen and of weak fluctuations. We make our estimates for the following characteristic inhomogeneity types: 1) the isotropic Kolmogorov turbulence and 2) the anisotropic saturated internal gravity waves. We obtain the expressions for the variances of the amplitude and phase fluctuations of radio occultation signals, as well as their estimates for the typical parameters of inhomogeneity models. From the GPS/MET observations, we evaluate the spectra of the amplitude and phase fluctuations in the altitude interval from 4 to 25 km in the middle and polar latitudes. As indicated by theoretical and experimental estimates, the main contribution into the radio signal fluctuations comes from the internal gravity waves. The influence of the Kolmogorov turbulence is negligible. We derive simple relationships that link the parameters of internal gravity waves and the statistical characteristics of the radio signal fluctuations. These results may serve as the basis for the global monitoring of the wave activity in the stratosphere and upper troposphere.

## 1 Introduction

The regular radio occultation (RO) monitoring of the Earth's atmosphere was for the first time implemented with the aid of the low Earth orbiter (LEO) Microlab-1, which was equipped with a receiver of high-stable GPS signal at wavelengths of $\lambda_1 = 19.03$ cm and $\lambda_2 = 24.42$ cm at a sampling rate of 50 Hz (Ware et al., 1996). In processing RO observations, neutral atmospheric meteorological variables are retrieved from amplitude and phase measurements (Gorbunov and Lauritsen, 2004; Gorbunov et al., 2005; Gorbunov and Lauritsen, 2006), while the ionospheric contribution is removed by using the double-frequency linear combination at the same ray impact height (Vorob'ev and Krasil'nikova, 1994; Gorbunov, 2002b). The impressive success of the GPS/MET experiment stimulated further development of RO satellites and constellations, including CHAMP and COSMIC experiments. Currently, RO sounding is an important method of monitoring meteorological parameters of the Earth's atmosphere; RO data are assimilated by the world's leading numerical weather prediction centers (Rocken et al., 2000; Yunck et al., 2000; Steiner et al., 2001; Pingel and Rhodin, 2009; Poli et al., 2009; Cucurull, 2010; Poli et al., 2010; Rennie, 2010).

The stability of GPS signals, complemented with its global coverage and high vertical resolution, draws the attention of researchers to the study of inhomogeneities in atmospheric refractivity in addition to the retrieval of mean profiles (Belloul and Hauchecorne, 1997; Gurvich et al., 2000; Tsuda et al., 2000; Wang and Alexander, 2010; Cornman et al., 2004, 2012; Shume and Ao, 2016; Gubenko et al., 2008, 2011). Occultation-based methods of sounding atmospheric inhomogeneities have a long and successful history. Initially, they were used for sounding the atmospheres of other planets of the Solar system, using occultations of stars and artificial satellites (Yakovlev et al., 1974; Woo et al., 1980; Hubbard et al., 1988). For the Earth's atmosphere, occultation observations of stellar scintillations were performed at the orbital station Mir (Alexandrov et al., 1990; Gurvich et al., 2001a, b; Gurvich and Kan, 2003a, b). The observations of stellar scintillations indicated that the Earth's atmosphere is characterized by the following two types of inhomogeneities: 1) isotropic fluctuations and 2) strongly anisotropic layered structures. On the basis of these data, an empirical two-component model of 3D inhomogeneity spectrum was developed, the anisotropic component described by the model of saturated internal gravity waves (IGW), the isotropic component as Kolmogorov turbulence (Gurvich and Brekhovskikh, 2001; Gurvich and Kan, 2003a, b). The method of the retrieval of these parameters from the observations of stellar scintillations was successfully employed for the interpretation of the experimental data acquired at the Mir station. This method was further enhanced and applied for the bulk retrieval of IGW and turbulence parameters from the observations made by fast photometers at the GOMOS/ENVISAT satellite (Sofieva et al., 2007a). The retrievals are performed in the altitude range from 50–60 km down to 30 km (Sofieva et al., 2007b). The upper limit was determined by radiation shot noise, the lower limit was determined by the applicability condition of the Rytov weak fluctuation/scintillation theory.

In the radio band, the amplitude fluctuations are much smaller than in the visible band, therefore, the weak fluctuation theory may be applicable down to altitudes of several kilometers. The main limitation is due to the humidity fluctuations, whose role becomes significant in the troposphere. The upper boundary of the measurable fluctuation of RO signals is about 30–35 km where residual ionospheric fluctuations and measurement noise become dominant. Optical and radio monitoring of atmospheric inhomogeneities complement each other in the regard of their height ranges. For the visible band, stratospheric IGW and turbulence make approximately equal contributions intensity fluctuations (Gurvich and Kan, 2003a, b; Sofieva et al., 2007b). In the radio band, the leading cause of the inhomogeneities is saturated IGWs, whose spectra are characterized by a steep power spectral decrease with increasing wavenumber. Nowadays, an increasing number of papers discuss the use of GPS for the study of atmospheric inhomogeneities. Some papers link the fluctuations of the amplitude and the phase of radio signals in the stratosphere to IGWs (Tsuda et al., 2000; Steiner and Kirchengast, 2000; Wang and Alexander, 2010; Khaykin et al., 2015), while other papers attribute this part to isotropic turbulence in the lower stratosphere and troposphere (Cornman et al., 2004, 2012; Shume and Ao, 2016). Therefore, it is necessary to formulate clear criteria for determining what type of inhomogeneities, isotropic or anisotropic, dominate radio signal fluctuations.

The aims of this paper are to clarify the role of the two inhomogeneity types and to evaluate their actual contributions in the amplitude and phase of RO signals. Our analysis is based on the phase screen approximation and the weak fluctuation theory. In the framework of these approximations, we obtain simple analytical relationships for the variance of fluctuations of radio signals for anisotropic and isotropic inhomogeneities. At this stage of our study, we confined the analysis of experimental data to

height range from 25 down to 4 km in the middle and polar latitudes, in order to exclude the influence of complicated dynamics of lower-tropospheric humidity. Our aim is not a quantitative study of RO signal fluctuations, but rather a demonstration of the qualitative principal differences between the manifestations of turbulence and IGWs in RO signals. The paper is organized as follows. In Section 2, we consider the 3D models of anisotropic and isotropic atmospheric inhomogeneities, the phase screen approximation, the weak fluctuation theory, and the approximations entailed. In Section 3, we apply these methods to derive simple relationships for the statistical characteristics of RO signal fluctuations. In Section 4, we consider the experimental variances and fluctuation spectra of the amplitude and phase for the lower stratosphere and upper troposphere. In Section 5, we discuss the relative contributions to RO signal fluctuations of isotropic and anisotropic inhomogeneities. In Section 6, we offer our conclusions.

## 2   Basic Models and Approximation

For RO signal analysis, we employ the following approximations:

1) a two-component model of the 3D spectrum of the atmospheric refractivity fluctuations;

2) the approximation of the equivalent phase screen;

3) the first order approximation of the weak fluctuation (the Rytov approximation).

### 2.1   3D Models of Refractivity Fluctuation Spectra

For the description of the wave propagation, we define the characteristics of the random media by its 3D spectrum of the relative fluctuations of refractivity $\nu = (N - \bar{N})/\bar{N}$, where $N = n - 1$, $n$ is the refractive index, and the overbar denotes the regional and seasonal statistical average estimate. The planetary waves and synoptic disturbances have much larger spatial and temporal scales significantly compared to the characteristic scales of RO signal fluctuations, including the Fresnel zone, the outer and inner scales of the inhomogeneities. This allows disregarding the large-scale processes. We assume the regular atmosphere, $\bar{N}$, to be locally spherically symmetric. In the visible band, refractivity fluctuations depend only on temperature fluctuations. In the radio range, humidity fluctuations make an additional contribution into refractivity fluctuations, which may be crucial in the lower troposphere (Eaton et al., 1988).

Stellar occultations indicated that the atmosphere is characterized by two types of density fluctuations: 1) large-scale anisotropic ones and 2) isotropic ones (Gurvich and Kan, 2003a, b; Sofieva et al., 2007a). Based on these observations, Gurvich developed a 3D model of the spectrum of relative fluctuations of refractivity, which includes two statistically-independent components: 1) anisotropic fluctuations $\Phi_W$ and 2) isotropic fluctuations $\Phi_K$ (Gurvich and Brekhovskikh, 2001; Gurvich and Kan, 2003a, b):

$$\Phi_\nu(\boldsymbol{\kappa}) = \Phi_W(\boldsymbol{\kappa}) + \Phi_K(\kappa) \tag{1}$$

where $\boldsymbol{\kappa}$ is the 3D wave number. It is assumed that the random field $\nu$ is locally homogeneous on a sphere (Gurvich, 1984; Gurvich and Brekhovskikh, 2001). This allows taking the anisotropy of refractivity irregularities into account.

Both components of the spectrum have a power law interval with the power of $-\mu$, which is confined between the outer scale $L_{W,K}$ and the inner scale $l_{W,K}$ of the inhomogeneities. Both components can be expressed in the following general form:

$$\Phi = \Phi_{W,K} = AC_{W,K}^2 \eta^2 \left(\kappa_z^2 + \eta^2 \kappa_\perp^2 + K_{W,K}^2\right)^{-\mu/2} \phi\left(\frac{\kappa}{\kappa_{W,K}}\right),$$

$$\kappa^2 = \kappa_z^2 + \eta^2 \kappa_\perp^2, \quad \kappa_\perp^2 = \kappa_x^2 + \kappa_y^2 \tag{2}$$

where $C_{W,K}^2$ are the structure constants determining the fluctuation intensity $\nu$, $\eta \geq 1$ is the anisotropy coefficient defined as the ratio of the characteristic horizontal and vertical scales, $\kappa_z$ is the vertical wavenumber, $\kappa_x$, $\kappa_y$ are the horizontal wavenumbers, the direction of axis $x$ coincides with that of the incident ray, $K_{W,K} = 2\pi/L_{W,K}$ and $k_{W,K} = 2\pi/l_{W,K}$ are wavevector parameters corresponding to the outer and inner scales, respectively. The function $\phi$ determines the damping of the spectrum for the smallest scales. We will use the following function: $\phi = \exp\left(-\kappa^2/\kappa_{W,K}^2\right)$. In our model, $C_{W,K}^2$, $\eta$, $K_{W,K}$ and $k_{W,K}$ are considered independent parameters.

For $\mu = 5$, $\eta \gg 1$, $A = 1$ the spectrum (2) $\Phi = \Phi_W$ is a 3D generalization of the known model of saturated IGWs with the vertical 1D spectrum with the slope $-3$ referred to as the "universal spectrum" (Dewan and Good, 1986; Smith et al., 1987; Fritts, 1989). We will use a model of the IGW spectrum with a constant anisotropy $\eta = \text{const} \gg 1$, although the latest studies of stellar scintillations (Kan et al., 2012, 2014) indicate that the anisotropy increases and saturates with increasing scale; the saturation value being about 100 for vertical scales of about 100 m. We will show that RO signal fluctuations are determined primarily by the Fresnel scale $\rho_F$ and the outer scale $L$. For radio waves with $\lambda = 20$ cm at a GPS–LEO path, $\rho_F$ equals about 1 km, while the vertical outer scale $L_W$ is several kilometers. For inhomogeneities with scales $\geq 1$ km, the anisotropy $\eta$ significantly exceeds its critical value $\eta_{cr} = \sqrt{R_e/H_0} \approx 30$, where $R_e$ is the Earth's radius, and $H_0 = 6$–8 km is the atmospheric scale height (Gurvich and Brekhovskikh, 2001). Due to along-track curvature of the Earth, different orientations of anisotropic layered inhomogeneities with respect the line of sight result in saturation of eikonal, or phase fluctuations at $\eta \approx \eta_{cr}$. This is explained by the fact that for inhomogeneities inclined with respect to the line of sight, a ray is only influenced by a limited horizontal piece of each inhomogeneity. For a larger anisotropy $\eta \gg \eta_{cr}$, their dependence on $\eta$ degrades, and they remain at the value corresponding to the asymptotic case of spherically-layered inhomogeneities (Gurvich, 1984; Gurvich and Brekhovskikh, 2001). In more detail, the concept of the critical anisotropy $\eta_{cr}$ will be ducussed below (see Eqs. (7) and (8)). Therefore, for RO sounding, the approximation of strongly anisotropic IGW inhomogeneities $\eta = \text{const} \gg \eta_{cr}$ is acceptable. The structure characteristic for dry air $C_{W,dry}^2$ is expressed in terms of the conventional parameters determining the 1D vertical spectrum of temperature fluctuations in the IGW model, $V_{\delta T/T}(\kappa_z) = \beta \frac{\omega_{B.V.}^4}{g^2}\kappa_z^{-3}$ (Smith et al., 1987; Fritts, 1989; Tsuda et al., 1991), as follows (Sofieva et al., 2009):

$$C_{W,dry}^2 = \frac{3\beta\omega_{B.V.}^4}{4\pi g^2} \tag{3}$$

where $\beta \approx 0.1$ is the coefficient introduced in the IGW model, $\omega_{B.V.}$ is the Brunt–Väisälä frequency, and $g$ is the gravitational acceleration. More discussion on the model $\Phi_W$ is presented below in Section 5.

To obtain the value of the structure characteristic $C_W^2$ in the radio band, $C_{W,dry}^2$ must be multiplied with the coefficient $K^2$, which takes humidity into account (Tatarskii, 1971; Good et al., 1982; Tsuda et al., 2000). The inner scale $l_W$ may vary

in the stratosphere from several meters to several tens of meters (Gurvich and Kan, 2003b; Sofieva et al., 2007a). For locally homogeneous random fields, the exponent $\mu$ of a purely power-law spectrum must lie between 3 and 5 (Rytov et al., 1989a, b). This dictates the necessity of introduction of the outer scale, although the variance of amplitude fluctuations only indicates a weak dependence from the outer scales up to $\mu < 6$ (Gurvich and Brekhovskikh, 2001).

For $\mu = 11/3$, $\eta = 1$, $A = 0.033$, and $C_K^2 = C_n^2 / \bar{N}^2$, where $C_n^2$ is the structure characteristic of refractivity fluctuations, and the spectrum (2) $\Phi = \Phi_K$ is a model of the Kolmogorov isotropic turbulence (Monin and Yaglom, 1975). In a stably stratified atmosphere, turbulence is developed mostly in separate layers with vertical scales from several tens of meters to one kilometer. We use the characteristic scale of these layers as the estimate of the outer scale of isotropic turbulence. The inner scale in the spectrum of the Kolmogorov turbulence can be defined as $l_K \approx 6.5\lambda_K = 6.5\nu_a^{3/4}\varepsilon_k^{-1/4}$, where $\lambda_K$ is the Kolmogorov scale,
$\nu_a$ is the kinematic molecular viscosity, $\varepsilon_k$ is the kinetic energy dissipation rate (Tatarskii, 1971).

## 2.2   Approximations of Phase Screen and Weak Fluctuations

Due to the exponential decay of air density with the altitude, a ray propagating in the atmosphere is mainly affected by the vicinity of the ray perigee, with the effective size along the ray of about several hundred kilometers. The distance from the perigee to the LEO is much greater, about 3000 km. This allows the approximation of the atmosphere as a thin screen that
only introduces phase variations, including both regular and random ones, and is referred to as a phase screen. The amplitude fluctuations are formed due to the diffraction during the propagation in the free space from the screen to the receiver. We position the phase screen in the plane crossing the Earth's center and perpendicular to the incident rays. The occultation geometry has been discussed in many papers: (Vorob'ev and Krasil'nikova, 1994; Ware et al., 1996; Gorbunov and Lauritsen, 2004; Cornman et al., 2004; Pavelyev et al., 2012, see further references and Figures therein). The phase screen has been
discussed in (Hubbard et al., 1978; Woo et al., 1980; Gurvich, 1984; Gurvich and Brekhovskikh, 2001). The use of the phase screen allows a significant simplification of the RO signal fluctuation analysis, and makes it possible to take into account the regular variation of refraction with altitude. Only when evaluating the phase shift (eikonal) is it necessary to take the Earth's curvature into consideration.

     The amplitude fluctuations are considered weak if their variance is less than unity (Tatarskii, 1971; Ishimaru, 1978). The
weak fluctuation approximation makes it possible to derive simple linear relationships linking the 3D spectrum of the atmospheric refractivity fluctuations with the 2D spectrum of amplitude and phase fluctuations of RO signal (Rytov et al., 1989a, b; Gurvich and Brekhovskikh, 2001; Sofieva et al., 2007a). In the visible range, fluctuations are weak for ray perigee altitudes above 25–30 km (Gurvich and Kan, 2003a, b; Sofieva et al., 2007b). For GPS radio signals, amplitude fluctuations are significantly weaker, because the Fresnel scale is about thousand times greater than that in the visible range. At low altitudes,
refractive attenuation also reduces amplitude fluctuations. Below, we will show that the weak fluctuation condition for GPS RO observations can be fulfilled down to an altitude of several kilometers. In the lower troposphere, especially in the tropics, the influence of humidity is strong, and amplitude fluctuations may become strong due to multipath propagation. Complicated non-linear relationships for strong fluctuations may significantly aggravate the data analysis. Some of options of the retrieval of inhomogeneity parameters under strong fluctuation conditions are discussed, for example, by Gurvich et al. (2006).

Because the velocity of the ray immersion in satellite observations is large compared to the atmospheric motions associated with the refractivity inhomogeneities, it is possible to apply the hypothesis of "frozen" inhomogeneities for mapping measured temporal spectra of signal fluctuations into spatial spectra.

## 3    Relationships for Statistical Moment of RO Signal Parameters

The approximations of phase screen and weak fluctuations allow deriving simple expressions for the statistical moments of RO signal fluctuations. In this Section, we will discuss the relationships that link the fluctuations of RO signals with those of atmospheric refractivity for IGW and turbulence models, as well as the mean profiles of variances of RO signal fluctuations.

### 3.1    Correlation Functions and Spectra

For a satellite-to-satellite path, using the approximations of phase screen and weak fluctuation, it is possible to derive the
following 2D correlation functions in the observation plane $(z_0, y_0)$ (Rytov et al., 1989b):

$$B_{\chi,S}(\Delta z_0, \Delta y_0) =$$
$$= \frac{1}{2} \left\{ \tilde{B}_S(\Delta z, \Delta y) \mp \frac{k\gamma}{4\pi x_1 q^{1/2}} \iint \tilde{B}_S(\Delta z', \Delta y') \sin\left[\frac{k\gamma}{4x_1 q}(\Delta z' - \Delta z)^2 + \frac{k\gamma}{4x_1}(\Delta y' - \Delta y)^2\right] d\Delta z' d\Delta y' \right\}$$

$$B_{\chi S}(\Delta z_0, \Delta y_0) =$$
$$= \frac{1}{2}\frac{k\gamma}{4\pi x_1 q^{1/2}} \iint \tilde{B}_S(\Delta z', \Delta y') \cos\left[\frac{k\gamma}{4x_1 q}(\Delta z' - \Delta z)^2 + \frac{k\gamma}{4x_1}(\Delta y' - \Delta y)^2\right] d\Delta z' d\Delta y' \tag{4}$$

where $\chi$ is the logarithmic amplitude, $S$ is the phase, $k = 2\pi/\lambda$, axis $x_0$ is collinear with the incident ray direction, axis $y_0$ is transverse, and axis $z_0$ is vertical, $\gamma = \frac{x_t + x_1}{x_t}$, $x_t$ is the distance from the transmitter to the phase screen, $x_1$ is the distance from the phase screen to the receiver, $q$ is refractive attenuation coefficient, $\Delta z, \Delta y$ are the scales in the phase screen, defined as the coordinate differences of the phase stationary points, and linked to the corresponding scales in the observation plane by the following relationships: $\Delta z = \frac{q}{\gamma}\Delta z_0, \Delta y = \frac{1}{\gamma}\Delta y_0$, $\tilde{B}_S(\Delta z, \Delta y)$ is the correlation function of the phase in the phase screen,
$B_{\chi S}$ is the mutual correlation function of the logarithmic amplitude and phase. The negative sign in the upper formula in (4) applies to the amplitude, and the positive sign applies to the phase.

Taking the Fourier transform, we arrive at the following expressions for the 2D fluctuation spectra of the received signal:

$$F_{\chi,S}(\kappa_z, \kappa_y) = \frac{k^2}{2}\left\{1 \mp \cos\left[\frac{x_1 q}{k\gamma}(\kappa_z^2 + q^{-1}\kappa_y^2)\right] \tilde{F}_\varphi(\kappa_z, \kappa_y)\right\}$$

$$F_{\chi S}(\kappa_z, \kappa_y) = \frac{k^2}{2}\sin\left[\frac{x_1 q}{k\gamma}(\kappa_z^2 + q^{-1}\kappa_y^2)\right] \tilde{F}_\varphi(\kappa_z, \kappa_y) \tag{5}$$

where $\tilde{F}_\varphi(\kappa_z, \kappa_y)$ is the 2D spectrum of the fluctuation of eikonal $\varphi = S/k$ in the phase screen. For the sake of convenience, relationships (5) are written in terms of wavenumbers $\kappa_z, \kappa_y$ in the phase screen, which are linked to the wavenumber in the observation plane by the inverse scale relations.

In the general case, the relationship between the 2D spectrum of the eikonal fluctuations in the phase screen $\tilde{F}_\varphi$ and 3D spectrum of the atmospheric refractivity fluctuations $\Phi$ for a random field $\nu$ that is locally homogeneous in a spherical layer can be written down as follows (Gurvich, 1984; Gurvich and Brekhovskikh, 2001):

$$\tilde{F}_\varphi(\kappa_z, \kappa_y) = \bar{\Psi}^2 \int \Phi\left(\kappa_z, \sqrt{\kappa_x^2 + \kappa_y^2}\right) \exp\left(-\frac{R_e H_0}{1 + \kappa_z^2 H_0^2}\kappa_x^2\right) \frac{d\kappa_x}{\sqrt{1 + \kappa_z^2 H_0^2}} \tag{6}$$

5 where $\bar{\Psi}$ is the mean eikonal. In particular, for the exponential atmosphere $\bar{\Psi} = \sqrt{2\pi R_e H_0}\bar{N}$. The eikonal, or the optical path, characterizes the propagation media, while the phase also depends on wavelength. In the RO terminology, the excess phase (or phase excess) refers to the eikonal of the observed field with the subtraction of the satellite-to-satellite distance. The excess phase, therefore, characterizes the atmospheric effect in the observed eikonal. The excess phase (eikonal) is modeled by the phase screen. Accordingly, in the observation plane we study the fluctuations for both eikonal and phase.

10 Formula (6) takes into account the along-track curvature of the Earth, which is essential, if $\eta \geq \eta_{cr}$. Figure 1 in (Gurvich and Brekhovskikh, 2001) provides a good illustration of the influence of the curvature upon the eikonal fluctuations in sounding isotropic and anisotropic atmospheric inhomogeneities. A general expression (6) for $\tilde{F}_\varphi$ is derived in (Gurvich and Brekhovskikh, 2001). Here, we will discuss important particular cases.

1. Moderate anisotropy $1 \leq \eta \ll \eta_{cr} \approx 30$. In this case the along-track curvature of the Earth is insignificant, and, assuming 15 $R_e H_0 \to \infty$ and performing integration, we arrive at the following known relationship, applicable for random inhomogeneities, locally homogeneous in the Cartesian coordinate system (Tatarskii, 1971; Rytov et al., 1989b), which we denote $\tilde{F}_\varphi^i$:

$$\tilde{F}_\varphi^i(\kappa_z, \kappa_y) \approx \bar{\Psi}^2 \sqrt{\frac{\pi}{R_e H_0}}\Phi(\kappa_z, \kappa_y, 0) \tag{7}$$

For the isotropic turbulence, we substitute $\Phi = \Phi_K$ with $\mu = 11/3$ and $\eta = 1$.

2. Strongly anisotropic inhomogeneities $\eta \gg \eta_{cr}$. For $\mu=5$, this case corresponds to the model of saturated IGW, for the 20 large-scale part of the spectrum. In this case, we can write the following expression for the eikonal spectrum $\tilde{F}_\varphi^a$:

$$\tilde{F}_\varphi^a(\kappa_z, \kappa_y) \approx \bar{\Psi}^2 \sqrt{\frac{\pi}{1 + \kappa_z^2 H_0^2}} \frac{\left(K_W^2 + \kappa_z^2 + \eta^2\kappa_y^2\right)^{1/2}\Gamma\left(\frac{\mu-1}{2}\right)}{\eta\Gamma\left(\frac{\mu}{2}\right)}\Phi_W(\kappa_z, \kappa_y, 0) \tag{8}$$

For strongly anisotropic inhomogeneities, function $\tilde{F}_\varphi^a$ has a sharp peak with respect to its argument $\kappa_y$, and it only differs from 0 in a small area near $\kappa_y = 0$; this corresponds to the asymptotic case of spherically symmetric inhomogeneities. This function can thus be approximated as $\tilde{F}_\varphi^a(\kappa_z, \kappa_y) \approx \tilde{V}_\varphi^a(\kappa_z)\delta(\kappa_y)$, where $\tilde{V}_\varphi^a(\kappa_z)$ is the 1D vertical spectrum of the eikonal 25 in the phase screen, and $\tilde{V}_\varphi^a(\kappa_z)$ is evaluated by integrating (8) with respect to horizontal wavenumbers:

$$\tilde{V}_\varphi^a(\kappa_z) = \int \tilde{F}_\varphi^a(\kappa_z, \kappa_y)\, d\kappa_y =$$
$$= \bar{\Psi}^2 C_W^2 \sqrt{\frac{\pi}{1 + \kappa_z^2 H_0^2}}\frac{\Gamma\left(\frac{\mu-1}{2}\right)}{\eta\Gamma\left(\frac{\mu}{2}\right)}\exp\left(-\frac{\kappa_z^2}{\kappa_W^2}\right)\left(K_W^2 + \kappa_z^2\right)^{-\frac{\mu}{2}+1}\Gamma\left(\frac{1}{2}\right)U\left(\frac{1}{2}, -\frac{\mu-4}{2}, \frac{K_W^2 + \kappa_z^2}{\kappa_W^2}\right) \tag{9}$$

where $U(\alpha, \beta; t)$ denotes the hypergeometric function.

The variance of the logarithmic amplitude fluctuation is determined by the scales of the order of the Fresnel zone (Tatarskii, 1971). The vertical Fresnel scale $\rho_F = \sqrt{\pi \lambda x_1 q / \gamma}$ for $\lambda = 19.03$ cm varies from 1260 m at a ray height of about 30 km to about 500 m at a ray perigee height of 2 km; therefore, in our case $\rho_F \gg l_W$. The variances of eikonal and refraction angle fluctuations, and the mutual correlation function of amplitude and phase for such a steep 3D spectrum ($\mu = 5$) are determined by scales of the order of the outer scale $L_W$. Therefore, for the IGW model, the fluctuations of all the RO signal parameters under discussion are determined by inhomogeneities with relatives large vertical scales, significantly exceeding the inner scale: $\kappa_z \ll \kappa_W$. Then, using the expansion of the hypergeometric function for small arguments $t$, it is possible to derive the following expression (Gurvich, 1984):

$$\tilde{V}_\varphi^a (\kappa_z) \approx 2\pi \bar{\Psi}^2 C_W^2 \frac{\left(\mathrm{K}_W^2 + \kappa_z^2\right)^{-\frac{\mu}{2}+1}}{(\mu - 2)\sqrt{1 + \kappa_z^2 H_0^2}} \tag{10}$$

In this case, the vertical fluctuation spectrum of $\nu$, which corresponds to relative temperature fluctuations $\delta T / \bar{T}$ for dry atmosphere, $V_W^a (\kappa_z)$, and the eikonal fluctuation spectrum $\tilde{V}_\varphi^a (\kappa_z) / \bar{\Psi}^2$ in the phase screen are linked by the following relationship (Gurvich, 1984):

$$V_W^a (\kappa_z) = 4\pi C_W^2 \frac{\left(\mathrm{K}_W^2 + \kappa_z^2\right)^{-\frac{\mu}{2}+1}}{(\mu - 2)} = \sqrt{1 + \kappa_z^2 H_0^2} \frac{\tilde{V}_\varphi^a (\kappa_z)}{\bar{\Psi}^2} \tag{11}$$

Relationship (11) is written for single-sided spectra for $\kappa_z \geq 0$.

In the observations, we obtain a 1D realization of the signal along the receiver trajectory. During a RO event, the changes of the satellite positions are small with respect to their distance from the phase screen. Moreover, the fluctuation correlation scale along the ray significantly exceeds the correlation scale in the transverse direction (Tatarskii, 1971). Therefore, a measured realization corresponds to the ray displacement in the phase screen by distance $s$ along the projection of the satellite trajectory arc. In the phase screen model, we have to take into account the refractive deceleration of ray immersion, and the vertical compaction of the scales. The observation geometry will be determined by the obliquity angle $\alpha$ of the occultation plane, defined as the angle between the immersion direction of the ray perigee and the local vertical in the phase screen. 1D spectra of amplitude and phase fluctuation measured along arc $s$ at angle $\alpha$ can be expressed as follows (Gurvich and Brekhovskikh, 2001):

$$V_{\chi,S} (\kappa_s) = \int F_{\chi,S} \left(\kappa_s \sin\alpha + \kappa' \cos\alpha, \kappa_s \cos\alpha - \kappa' \sin\alpha\right) d\kappa' \tag{12}$$

Angle $\alpha = 0°$ corresponds to a vertical occultation, and $\alpha = 90°$ corresponds to a horizontal, or grazing occultation. The frequency of amplitude fluctuations $f$ is linked to the wavenumber $\kappa_s$ by the following relationship:

$$\kappa_s = 2\pi f / v_s \tag{13}$$

where $v_s$ is the velocity of the ray perigee projection to the phase screen.

For isotropic inhomogeneities, the characteristic frequencies are determined by the corresponding scales and oblique velocity $v_s$ in the direction at angle $\alpha$. Strongly anisotropic inhomogeneities are intersected by the line of sight, effectively, in the

vertical direction, because the effect of the horizontal velocity component is much smaller. The condition of such effectively vertical occultations is as follows (Kan, 2004): $\tan \alpha < \eta$. For $\eta \geq 50$, this condition is fulfilled up to angles $\alpha \approx 89°$, which applies, eventually, to any occultation. For strongly anisotropic inhomogeneities, the 1D vertical spectra of amplitude and phase fluctuations at the receiver, are described by the following simple relationships:

$$V_{\chi,S}^{a}(\kappa_z) = \frac{k^2}{2} \left[ 1 \mp \cos\left( \frac{x_1 q}{k\gamma} \kappa_z^2 \right) \right] \tilde{V}_{\varphi}^{a}(\kappa_z) \tag{14}$$

where $\tilde{V}_{\varphi}^{a}(\kappa_z)$ is determined by (10). As above, the negative sign applies to the amplitude, and the positive sign applies to the phase.

## 3.2 Variance of Logarithmic Amplitude Fluctuations

For the fluctuations of logarithmic amplitude, in both models of the 3D spectrum of inhomogeneities, the principle scale is the
Fresnel scale $\rho_F$, which significantly exceeds the inner scale. In addition, assuming that the outer scale is much greater than $\rho_F$, we can omit the outer and inner scale in Eq. (2) for the 3D spectrum and use it as a pure power law.

In the case of strongly anisotropic inhomogeneities $\eta \gg \eta_{cr}$, using (10), (14), and condition $\kappa_z H_0 \gg 1$, we can derive the following expression for the variance of logarithmic amplitude (Gurvich, 1984):

$$\sigma_{\chi}^2 (\eta \gg \eta_{cr}) \approx \frac{k^2}{2} \iint \left\{ 1 - \cos\left[ \frac{x_1 q}{k\gamma} \left( \kappa_z^2 + q^{-1}\kappa_y^2 \right) \right] \right\} \tilde{V}_{\varphi}^{a}(\kappa_z) \delta(\kappa_y) d\kappa_y d\kappa_z =$$

$$\qquad = \frac{\pi^2 C_W^2 \bar{\Psi}^2 k^2}{2 H_0 (\mu - 2) \Gamma\left(\frac{\mu}{2}\right) \sin\left(\pi \frac{\mu-2}{4}\right)} \left( \frac{q z_1}{\gamma k} \right)^{\frac{\mu}{2} - 1} \tag{15}$$

which, for $\mu = 5$, correspond to the model of saturated IGWs.

For a moderate anisotropy $1 \leq \eta \ll \eta_{cr}$, the corresponding expression can also be found in (Gurvich, 1984):

$$\sigma_{\chi}^2 (\eta \ll \eta_{cr}) = \frac{k^2}{2} \iint \left\{ 1 - \cos\left[ \frac{x_1 q}{k\gamma} \left( \kappa_z^2 + q^{-1}\kappa_y^2 \right) \right] \right\} \tilde{F}_{\varphi}^{i}(\kappa_y, \kappa_z) d\kappa_y d\kappa_z =$$

$$\qquad\qquad = \frac{A \pi^2 \sqrt{\pi} C^2 \bar{\Psi}^2 k^2 \eta}{4 \sqrt{R_e H_0} \Gamma\left(\frac{\mu}{2}\right) \sin\left(\pi \frac{\mu-2}{4}\right)} \left( \frac{q z_1}{\gamma k} \right)^{\frac{\mu}{2} - 1} \tag{16}$$

For $C^2 = C_K^2$, $\mu = 11/3$, $A = 0.033$, and $\eta = 1$, Eq. (16) corresponds to a locally homogeneous turbulence.

In order to analyze the influence of anisotropy upon amplitude fluctuations, consider the ratio of (15) and (16) with the same power exponent $\mu$:

$$\frac{\sigma_{\chi}^2 (\eta \gg \eta_{cr})}{\sigma_{\chi}^2 (\eta \ll \eta_{cr})} = \frac{2}{\sqrt{\pi} (\mu - 2)} \frac{\eta_{cr}}{\eta} \tag{17}$$

For $\eta = 1$ and $\mu = 5$, this ratio equals 12, and for $\mu = 11/3$, it equals 20. Therefore, the variance of logarithmic amplitude
fluctuations increases with increasing anisotropy $\eta$ for $\eta < \eta_{cr}$, and saturates for $\eta \approx \eta_{cr}$, and for the extreme case of spherically layered inhomogeneities $\eta \approx 100 \gg \eta_{cr}$, the ratio in question is about 10–20. This is a consequence of the geometry of occultations: rays are oriented lengthwise with respect to prolonged inhomogeneities.

### 3.3 Variance of Phase (Eikonal) Fluctuations

The main contribution into phase fluctuations comes from inhomogeneities with vertical scales close to the outer scale. Therefore, it is possible to use the geometric optical approximation for formulas (5) and (14). To this end, we expand the cosine for small arguments into series and neglect the inner scale.

For the variance of phase fluctuations for strong anisotropy $\eta \gg \eta_{cr}$ and outer scale $K_W^{-1} \approx H_0/2\pi$, we obtain:

$$\sigma_S^2 \left( \eta \gg \eta_{cr} \right) \approx \frac{2\pi \sqrt{\pi} k^2 C_W^2 \bar{\Psi}^2 \Gamma \left( \frac{\mu - 2}{2} \right)}{(\mu - 2) H_0 \Gamma \left( \frac{\mu - 1}{2} \right)} K_W^{-\mu + 2} \tag{18}$$

For $\mu = 5$, the variance depends on the outer scale as $K_W^{-3}$.

     For a moderate anisotropy $\eta \ll \eta_{cr}$, using (5) and (7), we obtain the following expression:

$$\sigma_S^2 \left( \eta \ll \eta_{cr} \right) \approx \frac{2\pi \sqrt{\pi} k^2 A C^2 \bar{\Psi}^2 \eta}{(\mu - 2) \sqrt{R_e H_0}} K_K^{-\mu + 2} \tag{19}$$

For $C^2 = C_K^2$, $\mu = 11/3$, $A = 0.033$, and $\eta = 1$, this corresponds to the model of isotropic turbulence. In this case, the variance of phase fluctuations depends on the outer scale as $K_K^{-5/3}$ (Tatarskii, 1971). The ratio of (18) and (19) for the same $\mu$, in way similar to amplitude fluctuations, is proportional to the ratio of $\eta_{cr}/\eta$.

### 3.4 Variance of Ray Incident Angle Fluctuations

For a strong anisotropy $\eta \gg \eta_{cr}$, the incident ray direction fluctuations are nearly vertical. The vertical fluctuation spectrum of
the ray incident angle is equal to that of eikonal, multiplied by $\kappa_z^2$. Then, replacing the cosine in (14) by unity, we arrive at the following expression for the variance of ray incident angle fluctuations:

$$\sigma_\alpha^2 \left( \eta \gg \eta_{cr} \right) \approx \frac{4\pi C_W^2 \bar{\Psi}^2}{(\mu - 2)(\mu - 4) H_0} K_W^{-\mu + 4} \tag{20}$$

and for $\mu = 5$, the variance depends on the outer scale as $K_W^{-1}$.

     For the case $\eta = 1$, the variance of incident angle fluctuations is determined by the inner scale of inhomogeneities, unlike the
case of a strong anisotropy. Moreover, the term with the cosine in (5) gives a small contribution, as compared to 1 if $\frac{x_1 q \kappa_K^2}{k\gamma} \gg 1$. Using (5) and neglecting the cosine term, we arrive at the following expression for the fluctuations of the full incident angle $\theta$:

$$\sigma_\theta^2 \left( \eta = 1 \right) \approx \frac{\pi \sqrt{\pi} A C_K^2 \bar{\Psi}^2 \Gamma \left( 2 - \frac{\mu}{2} \right)}{2 \sqrt{R_e H_0}} \kappa_K^{-\mu + 4} \tag{21}$$

For the Kolmogorov turbulence, the variance of incident angle fluctuations depends on the inner scale as $\kappa_K^{1/3}$. Introducing the effective thickness of the atmosphere along the ray, which equals $L_{ef} = \sqrt{\pi R_e H_0} \approx 400$ km, we see that (21) coincides with
the corresponding formula in (Tatarskii, 1971) for an observation distance of $L_{ef}$ in a homogeneously random medium.

### 3.5 Mutual Correlation of Logarithmic Amplitude and Phase

For the case of a strong anisotropy $\eta \gg \eta_{cr}$, the single-point correlation $\langle \chi S \rangle = B_{\chi S}(0)$ is determined by the outer scale of inhomogeneities. Using (5) and (10), and expanding the sine into series, we arrive at the following formula:

$$\langle \chi S (\eta \gg \eta_{cr}) \rangle = \frac{2\pi C_W^2 \bar{\Psi}^2 k^2}{H_0 (\mu - 2)(\mu - 4)} \frac{x_1 q}{k\gamma} \mathrm{K}_W^{-\mu+4} \tag{22}$$

For $\mu = 5$, the correlation depends on the outer scale as $\mathrm{K}_W^{-1}$, which is the same dependence as that of variance of bending angle fluctuations.

For isotropic inhomogeneities, $\eta = 1$, the most important scale determining the correlation of the logarithmic amplitude and phase, is the Fresnel scale $\rho_F \gg l_K$. Under the assumption that $\rho_F$ is small compared to the outer scale, it is sufficient to consider a 3D spectrum $\Phi_K$ in a purely power form. This results in the following formula:

$$\langle \chi S(\eta = 1) \rangle = \frac{\pi^2 \sqrt{\pi} A C_K^2 \bar{\Psi}^2 k^2}{4 \sqrt{R_e H_0} \Gamma\left(\frac{\mu}{2}\right) \cos\left(\pi \frac{\mu-2}{4}\right)} \left(\frac{x_1 q}{k\gamma}\right)^{\frac{\mu}{2}-1} = \sigma_\chi^2 (\eta = 1) \tan\left(\pi \frac{\mu-2}{4}\right) \tag{23}$$

For $\mu = 11/3$, the relation between correlation $\langle \chi S(\eta = 1) \rangle$ and the variance of amplitude fluctuations is the same as for a homogeneously random medium (Tatarskii, 1971).

### 3.6 Model Variance Profiles

The profiles were evaluated for a GPS–LEO system with orbit altitudes of 20000 km and 800 km, respectively, for a wavelength
of 19.03 cm. The parameters of the regular atmosphere, including refractive index $\bar{N}$, the height scale of a homogeneous atmosphere $H_0$, the average eikonal $\bar{\Psi}$, bending angle $\bar{\varepsilon}$, and refractive attenuation coefficient $q$, correspond to the standard model of the atmosphere.

The structure characteristic of the relative fluctuations of refractive index was specified for the model of saturated IGWs in a dry atmosphere, according to relation (3). Numerous radiosonde profiles and observations of stellar occultations indicate that
this relation is met with a good accuracy for the troposphere and stratosphere. For the radio band, the structure characteristic was multiplied by $K^2$ in order to take humidity into account. We were using the humidity profile, typical for spring and autumn in middle latitudes, with a scale height of 2.5 km. As shown above, the inner scale of the inhomogeneities is insignificant in the IGW model. The outer scale, corresponding to vertical scale of dominant waves, was assumed to equal $L_W = 4$ km (Smith et al., 1987; Tsuda et al., 1991).

For the isotropic turbulence at heights of 4–15 km, we used numerous data of radar measurements of the structure characteristic performed during years 1983–1984 in Platteville, Colorado (Nastrom et al., 1986). From the monthly averaged profiles of $C_n^2$ shown in Fig. 10 of the cited paper, we chose the maximum values that mostly correspond to August, in order to obtain the upper estimate of turbulent fluctuations of RO signals. For heights of 15–30 km, where humidity is negligible, we used model data $C_n^2$ from (Gracheva and Gurvich, 1980), which generalizes the results of numerous observations and models for
the optical turbulence (Gurvich et al., 1976), as well as retrievals of $C_n^2$ from stellar occultations (Gurvich and Kan, 2003b; Sofieva et al., 2007a). For the outer scale we used the largest possible value of 1 km, in order to obtain an upper estimate of the

isotropic turbulence contribution to RO signal fluctuations. The inner scale is assumed to increase from 4 cm at 4 km to 0.75 m at 30 km. The mean value of refraction angle $\bar{\varepsilon}$ and refractive attenuation coefficient $q$ were evaluated using the exponentially decaying atmospheric air density profile:

$$\bar{\varepsilon} = -\frac{\bar{\Psi}}{H_0}, \quad q = \left(1 - \frac{x_1}{\gamma}\frac{\bar{\varepsilon}}{H_0}\right)^{-1} \tag{24}$$

Figure 1 shows the model profiles of rms fluctuations of the logarithmic amplitude, eikonal, incident angle, as well as the correlation of the logarithmic amplitude and eikonal, for saturated IGWs and isotropic turbulence. For the assumed parameters of the 3D inhomogeneity spectra, all the rms for saturated IGWs exceed the corresponding values for the isotropic turbulence by an order of magnitude or even more, and this difference increases with the altitude. This especially applies to the eikonal fluctuations and amplitude-eikonal correlation, where the difference in rms exceeds two orders of magnitude. The knee of

the profiles at an altitude of 10 km for the turbulence model is linked to the peculiarity of the measured profiles of $C_n^2$ (Nastrom et al., 1986). In (Nastrom et al., 1986) it is, however, noted that the increase of $C_n^2$ above 10 km is not corroborated by other observations in Platteville (Ecklund et al., 1979) and can be attributed to measurement noise. For the IGW model, the knee is explained by the abrupt change of the Brunt-Väisälä frequency near the tropopause, according to (3).

For the turbulence model, we assumed the outer scale to be equal to 1 km. However, for stable stratification, which is typical

for the stratosphere, the outer scale may be significantly less than this value, down to hundreds or even tens of meters (Wheelon, 2004). The saturation of the 3D spectrum of turbulence at the outer scale, which is less than the Fresnel zone size, will result in the decrease of the turbulent fluctuations of RO signals as compared to the estimates for 1 km, and the difference with IGW fluctuations will be even larger. Due to the fact that the averaging over the Fresnel scale results in much smaller amplitude fluctuations of RO signal as compared to the visible band, the weak fluctuation condition is fulfilled down to the lower limit of

the altitude range under discussion.

It is known that local profiles of atmospheric inhomogeneities exhibit large natural variability. Furthermore, even their average profiles significantly vary depending on latitude, season, orography, and region. The turbulence structure characteristic for different observations, even in a free atmosphere, may vary by up to two orders of magnitude (e.g. Gracheva and Gurvich, 1980; Wheelon, 2004). Significant variability is observed for the intensity of saturated IGWs (e.g. Sofieva et al., 2007a, 2009),

which depends both on the sources producing the waves and on the propagation and breaking conditions. Eq. (3) for the saturated IGW only reflects the most general relation between the structure characteristic and the atmospheric stability. The latitudinal variability of the structure characteristic significantly exceeds that of $\omega_{B.V.}^4$ (Sofieva et al., 2009). However, on the average, the variations of refractivity fluctuations and, therefore, the amplitude fluctuations are determined by the exponential decay of the atmospheric density with altitude. Because our work is aimed at a qualitative distinction of the contribution of

turbulence and IGWs to the fluctuations of RO signals, we consider only averaged vertical profiles of the structure characteristic of turbulence and IGWs for the theoretical estimates. Quantitative studies of IGW parameters and wave activity for different latitudes, seasons, and regions in the stratosphere and upper troposphere are planned for the future work. Despite possible inaccuracies in the assumed values of the structure characteristic, the variance estimates obtained in this work, definitely indicate the dominant role of saturated IGWs under the conditions in question.

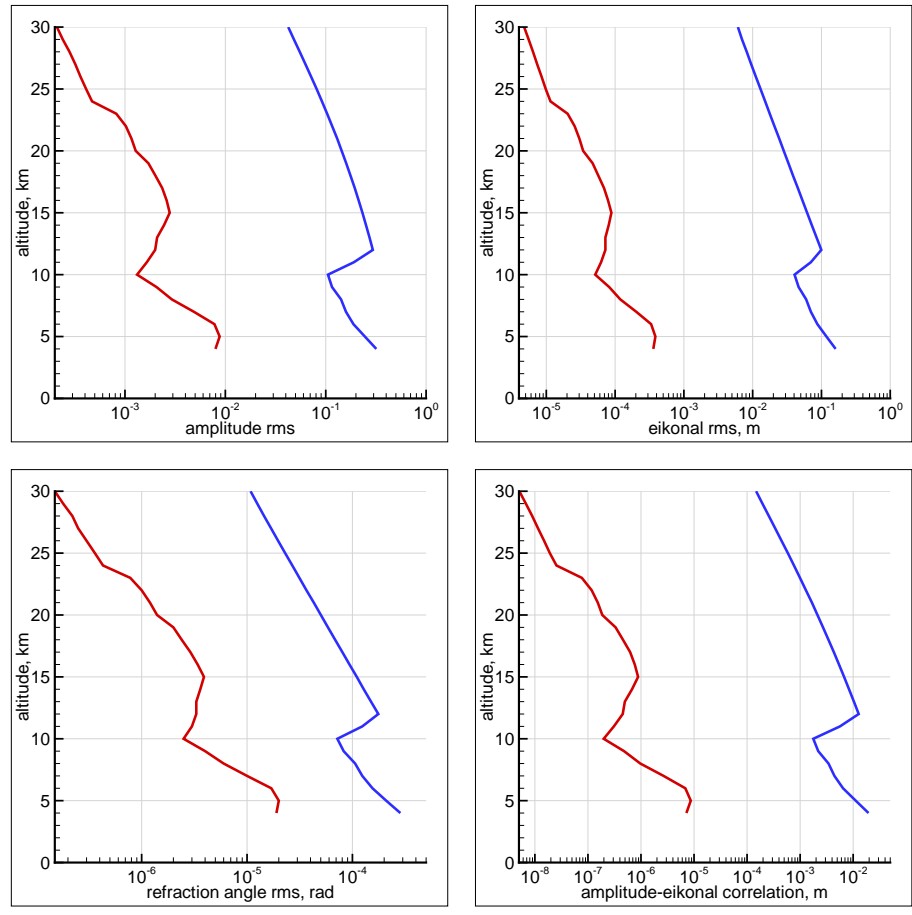

**Figure 1.** RMS fluctuations of logarithmic amplitude, eikonal, incident angle, and single-point correlation of logarithmic amplitude and phase for the model of saturated IGWs (blue lines) and for the model of turbulence (red lines).

## 4 Experimental Fluctuation Spectra of Amplitude and Phase

The most important difference between turbulence and saturated IGWs is the anisotropy of the latter. The variances of RO signal parameter fluctuations, being single-point characteristics, do not contain an immediate information on the anisotropy of the 2D field of RO signal fluctuation in the observation field. This information can be extracted from an ensemble of 1D spectra of RO signal fluctuations measured at different obliquity angles, when categorized according to frequency or to vertical wavenumber. For turbulence, due to its isotropy, the fluctuation frequencies $f$ of the signal are determined by the characteristic scales and the oblique movement velocity $v_s$ of the line of sight defined as the velocity of the projection of the ray perigee to the phase screen plane. For anisotropic IGW inhomogeneities, the fluctuation frequencies of the signal are determined by the vertical velocity $v_v$ for the majority of occultations. These velocities and frequencies coincide for a vertical occultation, but they may differ several tens of times for a highly oblique occultation. This frequency discrimination of isotropic and anisotropic fluctuations

in oblique occultations allows for a separate estimate of their contribution into signal fluctuation spectra (Gurvich and Kan, 2003a, b; Sofieva et al., 2007a, b).

Figures 2 and 3 show the spectra of relative fluctuations of the amplitude for the wavelength $\lambda_1 = 19.03$ cm from GPS/MET observations acquired on February 15, 1997. Figure 2 shows the spectra for the low stratosphere at altitudes from 25 km down to the upper boundary of the tropopause located at 9–13 km. Figure 3 shows the spectra for the upper troposphere at altitudes from 8–12 km down to 4 km. As noted above, the analysis is based on occultation events with different obliquity angles, in middle and polar latitudes. We selected events with a low level of ionospheric fluctuations at altitudes below 60–70 km. Under these conditions, below 25–30 km neutral atmospheric signal fluctuations will supersede the ionospheric fluctuations and measurement noise. Noise correction was performed under the assumption that the noise source is the receiver, and the noise properties remain constant during an occultation event. The noise spectrum was estimated from the occultation data records at altitudes of 70–50 km with a low level of neutral atmospheric and ionospheric fluctuations. The mean amplitude profiles were determined by linear fitting.. Figure 2 and 3 (as well as Figure 4 and 5 below) show 30 examples of stratospheric events and 20 examples of tropospheric events. For the stratosphere, the obliquity angles changed within the range of $20° − 87°$, for the troposphere they changed in the range of $35° − 88°$. Therefore, for strongly anisotropic inhomogeneities, these occultations were effectively vertical.

The amplitude fluctuation spectra are represented as the product of wavenumber and spectral density, normalized to the variance. Such a product will be hereinafter referred to as the spectrum, as distinct from the spectral density. The spectra indicate a maximum corresponding to the Fresnel scale. The theoretical spectra for both inhomogeneity types have asymptotics with a slope of $+1$ for low frequencies; for IGWs, the asymptotics corresponds to the condition $L_W \to \infty$. For the high frequencies, at the diffractive decline, the slope of the spectra is $-\mu + 2$, i.e. $-3$ for the IGW model, according to formulas (10) and (14), and $-5/3$ for turbulence (Tatarskii, 1971; Gurvich and Brekhovskikh, 2001; Woo et al., 1980). For the chosen fragments of realizations, we used the Hann cosine window. This window allows the minimization of distortions of spectra with a steep decrease (Bendat and Piersol, 1986). The Fourier periodograms were averaged with a spectra window with a variable width $\Delta f$: first with a window of a constant quality $f/\Delta f = 2$, then with a window of a constant width. The spectra were normalized to the variance, evaluated as the integral of the spectral density over frequency. The spectra in Figures 2 and 3 are plotted in two forms. In panels A, they are plotted as functions of the oblique wavenumber, according to the isotropy hypothesis; in panels B, they are plotted as functions of the vertical wavenumber, according to the anisotropy hypothesis for effectively vertical occultations. The ray perigee velocities and obliquity angles were evaluated from the satellite orbit data. The wavenumbers were normalized on the Fresnel scale in the corresponding direction, i.e., the values along the horizontal axis are $2\pi\kappa_s\rho_F(\alpha)$ for the isotropy hypothesis and $2\pi\kappa_z\rho_F(\alpha = 0)$ for the anisotropy hypothesis. For this normalization, the spectral maxima must correspond to the argument equal to 1.

Figure 2 and 3 indicate that for the isotropy hypothesis, panels A, the spectral maxima are spread over about 1.5 decade of frequencies. With the increasing obliquity angle, the maxima systematically shift to lower frequencies, although, the oblique velocities, on the contrary, increase. In panels B, all the spectra are peaked near wavenumber 1, which represents the first Fresnel zone. This favors the anisotropy hypothesis. For the verification of the isotropy/anisotropy hypotheses, strongly oblique

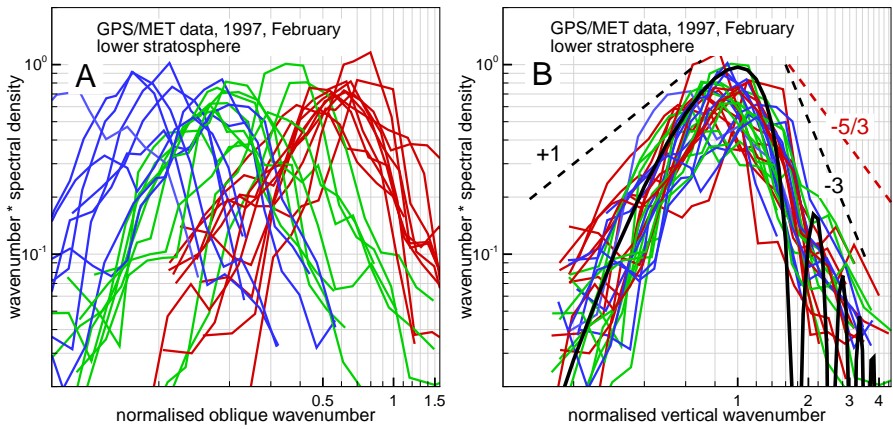

**Figure 2.** Amplitude fluctuation spectra for the lower stratosphere: panel A: spectra as functions of the oblique wavenumber corresponding to the isotropy hypothesis (Kolmogorov turbulence); panel B: spectra as functions of the vertical wavenumber corresponding to the anisotropy hypothesis (saturated IGWs). The color map red–green–blue corresponds to the increasing obliquity angles, which are subdivided into three groups. The black solid line in panel B presents the theoretical vertical spectrum for the saturated IGW model; the dashed lines present the asymptotics of this spectrum for low and high frequencies, the low-frequency asymptotic is evaluated for $L_W \to \infty$. For the comparison, red dashed lines show the high-frequency spectral asymptotic of the Kolmogorov turbulence model.

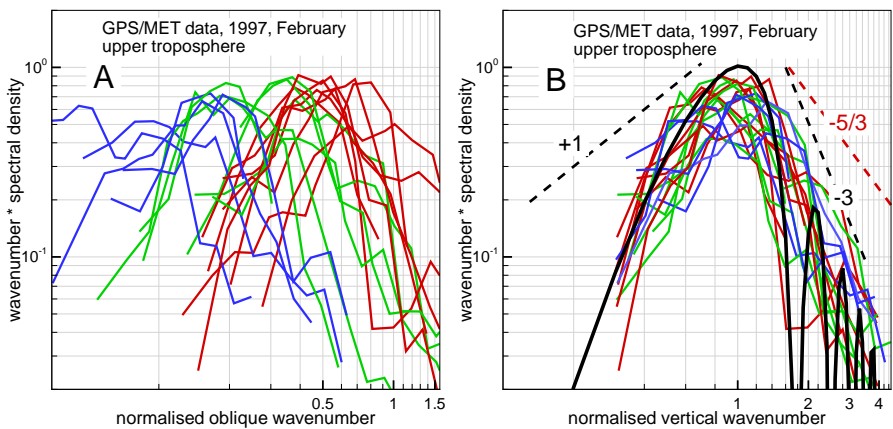

**Figure 3.** Amplitude fluctuation spectra for the upper troposphere: panel A: spectra as functions of the oblique wavenumber corresponding to the isotropy hypothesis (Kolmogorov turbulence); panel B: spectra as functions of the vertical wavenumber corresponding to the anisotropy hypothesis (saturated IGWs). The notations are the same as in Figure 2

occultations should be most informative. If the amplitude spectra contained a significant isotropic component, it should manifest itself in oblique spectra as an additional maximum at higher frequencies. In stellar scintillation spectra, such a double-hump structure is typical (Gurvich and Kan, 2003a, b; Sofieva et al., 2007a). The absence of the second high-frequency maximum in

Figures 2 and 3 indicates that the amplitude fluctuations caused by the isotropic turbulence in these measurements were significantly weaker compared to those caused by the anisotropic inhomogeneities. The experimental amplitude spectra in panels B are in a good agreement with the theoretical spectrum (10) and (14). The variance of amplitude fluctuations weakly depends on the outer scale $L_W$, if it significantly exceeds the Fresnel scale. Nevertheless, the influence of $L_W$ results in a faster than +1 decrease of the spectrum at low frequencies. This effect was utilized for the retrieval of internal gravity wave and turbulence parameters from stellar scintillations (Sofieva et al., 2007a). For the theoretical spectrum in the stratosphere, we used the value of $L_W = 2.0$ km; for the troposphere, we used the value of $L_W = 1.2$ km. The fringes of the theoretical spectrum in the high-frequency region are caused by diffraction on the phase screen. The slope of the spectrum at high frequencies agrees with the theoretical value $-3$; note, the diffractive slope of the spectral density equals $-4$. The fact that all the spectra in panels B group together means that all the obliquity angles $\alpha$ met the condition of effectively vertical occultations. For $\alpha_{max} = 88°$, we can estimate anisotropy $\eta > \tan(\alpha_{max}) \approx 30$ for the inhomogeneities, whose vertical scale equals the Fresnel scale.

The measured RMS values of the relative fluctuations of the amplitude in the stratosphere are 0.08–0.20, which is in a fair agreement with the IGW model (Figure 1), which equals 0.17 in the middle of the height range. For the upper troposphere, the measured RMS values are 0.12–0.35 and, therefore, they mostly exceed the theoretical estimate, which equals 0.16. The experimental RMS values proves the applicability of the approximation of weak fluctuations for the interpretation of these data.

The phase in RO observations is presented as the excess phase, which equals the difference between the full eikonal and the straight-line satellite-to-satellite distance. We will refer to the excess phase as the eikonal. Double-frequency observations allow for the exclusion of the ionospheric component of the eikonal under the assumption that the trajectories of the two rays coincide. The ionospheric corrected eikonal consists of two components: 1) the neutral atmospheric eikonal evaluated as the integral along a straight ray, which in Section 3 was denoted as $\Psi$, and 2) the addition to the geometrical length of the ray due to refraction (Vorob'ev and Krasil'nikova, 1984; Gurvich et al., 2000). The second component is approximately equal to the first one at a height of 15 km, and it rapidly increases for lower altitudes. In the first-order approximation of the perturbation method, the eikonal variations are determined by the refractive index variations of the neutral atmosphere (Vorob'ev and Krasil'nikova, 1984). Strong regular variations of the eikonal with the altitude, and its relatively small fluctuations, which amount to tenths of a percent, aggravate the separation of fluctuations, especially due to the difficulty of the evaluation of the mean eikonal profile (Gurvich et al., 2000; Cornman et al., 2012; Tsuda et al., 2000). In this study, we use the smooth eikonal profile evaluated for the MSISE-90 model (Hedin, 1991) complemented with a simple model of humidity: the relative humidity has a constant value of 80% below 15 km. These profiles are evaluated for the real observation geometry and they correctly represent both the atmospheric eikonal and the geometrical length of the ray. Still, both before and after the subtraction of the model profiles, the eikonal realizations contained low-frequency trends, due the model inaccuracy. We applied an additional detrending square-polynomial term to the eikonal deviation from the model. This procedure smooths the spectral components with scales exceeding the half-length of the realization. Similar to the amplitude spectra, we used the Hann cosine window also for the eikonal (Bendat and Piersol, 1986).

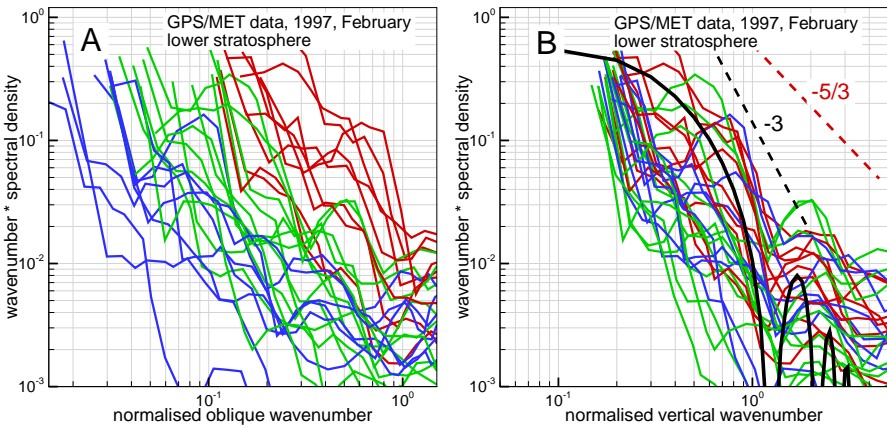

**Figure 4.** The normalized eikonal fluctuation spectra for the lower stratosphere: panel A: spectra as functions of the oblique wavenumber corresponding to the isotropy hypothesis; panel B: spectra as functions of the vertical wavenumber corresponding to the anisotropy hypothesis. The black solid line in panel B represents the theoretical vertical spectrum for the saturated IGW model; the dashed line represents its asymptotics. For the comparison, red dashed lines show the high-frequency spectral asymptotic of the Kolmogorov turbulence model. Cf. the caption of Figure 2.

Figures 4 and 5 present the normalized spectra of the atmospheric fluctuations of the eikonal for the same events and altitude ranges as for the amplitude spectra. The eikonal fluctuation spectra are also represented as the product of wavenumber and spectral density. In this representation, the slope of the theoretical eikonal spectra equals $-\mu+2$ and, correspondingly, it equals $-3$ for IGWs and $-5/3$ for turbulence. The eikonal spectra normalized according to the anisotropy hypothesis have a somewhat large spread compared to the amplitude spectra, still, they also corroborate the dominant role of anisotropic inhomogeneities. These spectra are in fair agreement with the theoretical vertical spectrum (14). For the evaluation of the theoretical spectrum, we used the same value of the outer scale as for the amplitude spectra.

For the stratosphere, the measured RMS values of the eikonal fluctuations are 3–10 cm, while their estimate was 5 cm. For the upper troposphere, the measured RMS were 4–15 cm, while their theoretical was about 7 cm.

The atmospheric inhomogeneity models have not only different anisotropy, but also different slope $-\mu$ of the 3D spectra, which determines the diffractive decay $-\mu+2$ in the presented spectra of RO amplitudes and phases. Due the fast decay, the noise limits the spectral range, which aggravates the derivation of accurate estimates.

Nevertheless, Figures 2-5 indicate that the diffractive decays of the experimental spectra are in a better agreement with the IGW model, as compared to the turbulence model.

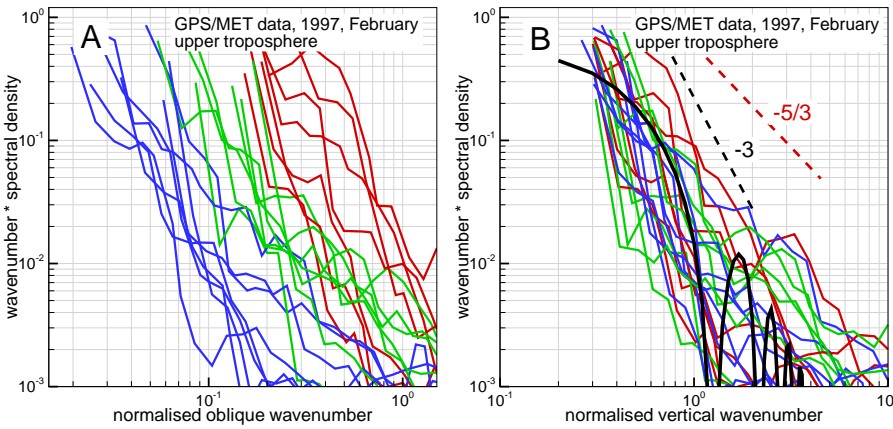

**Figure 5.** The normalized eikonal fluctuation spectra for the upper troposphere. Panel A: spectra as functions of the oblique wavenumber corresponding to the isotropy hypothesis; panel B: spectra as functions of the vertical wavenumber corresponding to the anisotropy hypothesis. The notations are the same as in Figure 4.

## 5 Discussion

In this study, we discussed the 3D spectra of atmospheric inhomogeneities of two types: 1) isotropic Kolmogorov turbulence, and 2) anisotropic saturated IGWs. For RO observations, in the approximations of the phase screen and weak fluctuations, we derived the relationships that link the observed 1D fluctuation spectra of the amplitude and phase with empirical 3D in-
5 homogeneity spectra. This allowed us to obtain the analytical expressions for the variances of the amplitude, phase, and ray incident angle fluctuations, as well as the single-point amplitude-phase correlation for both inhomogeneity types. The theoretical estimates of the variances of RO amplitude and phase fluctuations for different values of the parameters of atmospheric inhomogeneity model, including the structure characteristics and vertical scales, for middle latitudes in the stratosphere and upper troposphere, indicate that the major contribution into RO signal fluctuations comes from saturated IGWs. The contribu-
10 tion of the Kolmogorov turbulence, under these conditions, is small. Even taking into account a significant spread of possible values of the structure characteristics and typical scales of inhomogeneities, it is hard to expect that this can compensate the difference between the IGS and turbulence in this altitude range. Moreover, the averaging of RO signal fluctuations along the whole ray inside the atmosphere damps the influence of intermittence, which is typical for turbulence under stable stratification conditions.
15    For anisotropic inhomogeneities we employ an empirical model of saturated IGWs (2). Models of this type are widely used for the analysis of stellar and radio scintillations, the angular dependence of the back-scattering of radar signals, the retrieval of model parameters from occultations etc. 1D vertical and horizontal spectra of this model follow the $-3$ power. However, airborne observations (e.g. Nastrom and Gage, 1985; Bacmeister et al., 1996) indicate that the horizontal spectra of temperature fluctuations in the troposphere and stratosphere, have a power spectrum with a slope close to $-5/3$ in a wide range of scales

from several km to several hundred km (see also Dewan, 1994, the "saturated-cascade" model). In addition, the model (2) has a constant anisotropy. As noticed in section 2.1, observations of stellar occultations with grazing geometry (Kan et. al, 2014), together with the data about the anisotropy of dominant IGWs (e.g. Ern et al., 2004, the description of CRISTA experiment); GPS occultations in (Wang and Alexander, 2010) have revealed that the anisotropy coefficient is not uniform. It increases from
about 10–20 for the IGW breaking scale (10–20 m in the vertical direction) to the saturation value of several hundred for dominant IGWs.

The use of the simple model (2) for the problem in question is justified as follows. As shown above, the most important scales for the IGW model (the Fresnel scale and the outer scale), which determine the RO signal fluctuations, equal or exceed the value of about 1 km in the vertical direction. For inhomogeneities with such vertical scales, the anisotropy significantly
exceeds the critical value. Therefore, the amplitude and phase fluctuations do not any longer depend on the anisotropy values and reach the saturation level, as if the inhomogeneities were spherically symmetric. This explains why it is possible to use the model with a strong constant anisotropy. Due to this, the RO observation geometry can be assumed effectively vertical, and the amplitude and phase fluctuations only depend on the vertical structure of saturated IGWs (Eqs. (10) and (14)), which is adequately described by model (2). In some cases, for strongly oblique occultation events, the condition of effectively vertical
observation geometry may be broken, in the lowest few kilometers, due to the strong refraction, which decreases the vertical component of the ray immersion velocity.

Following the ideas of Dalaudier and Gurvich (1997), Gurvich and Chunchuzov (2008) developed an empirical 3D model of saturated IGWs, with the anisotropy increasing as a function of the vertical scale. The vertical spectrum follows the $-3$ power law, while the horizontal spectrum can have the $-5/3$ power law for the corresponding choice of anisotropy parameters.
This model is in a good agreement with the known air-borne observations of horizontal spectra of IGWs. Scintillation spectra evaluated on the basis of the variable anisotropy model (Gurvich and Chunchuzov, 2008) are in a good agreement with those evaluated on the basis of the constant anisotropy model (2) for effectively vertical occultations (Kan, 2016).

Joint observations of the amplitude and phase of RO signals open new pathways in the development and application of radio holographic methods. These methods allow enhancing the retrieval accuracy and resolution (e.g. Gorbunov and Gurvich,
1998a, b; Gorbunov, 2002a; Gorbunov and Lauritsen, 2004), as well as obtaining new information on the structure of the atmosphere (Pavelyev et al., 2012, 2015, and references therein). In particular, Pavelyev et al. (2015) demonstrated the potential of the locality principle for the localization and estimation of the parameters of layered structures, as well as the separation of the contributions of layered structures and turbulence in RO signals. In our study, we use the power spectra of the observed fluctuations of the amplitude and phase, correlated with the obliquity angle, in order to estimate and separate the contributions
of anisotropic inhomogeneities (saturated IGWs) and isotropic turbulence. The application of radio holographic methods for the enhancement of the accuracy and resolution is our plan for future work.

From GPS/MET data acquired on February 15, 1997, we evaluated the variances and spectra of the relative fluctuations of amplitude and the fluctuations of phase for the lower stratosphere, comprising the altitudes from 25 km down to the upper boundary of the tropopause, and for the upper troposphere, comprising the altitudes from the lower boundary of the tropopause
down to 4 km. For analysis, we chose RO events in middle and polar latitudes with different occultation trajectories: from ver-

tical ones with obliquity angle near 0 degrees to strongly oblique ones with obliquity angles up to 88 degrees. The experimental spectra of the amplitude and phase fluctuations, presented as a function of vertical wave numbers for the anisotropy hypothesis or oblique wavenumbers for the isotropy hypothesis, indicate strong anisotropy of the atmospheric inhomogeneities. This, along with the theoretical estimates signifies the dominant role of saturated IGWs for RO signal fluctuations. The experimental
estimates of variances of amplitude and phase fluctuations mostly agree with evaluations based on the IGW model.

In comparison with the visible band, the radio band is characterized by a much greater Fresnel scale $\rho_F$. This, together with the strong refractive attenuation at low altitudes, according to (15), significantly reduces the amplitude fluctuations and, therefore, the weak fluctuation condition is met for altitudes down to a few kilometers. This is corroborated by the measured variance of relative amplitude fluctuation. The upper boundary of the RO monitoring of atmospheric inhomogeneities is close to
the lower boundary of visible occultations. Therefore, radio and visible occultations, together with the simple approximations, permit a diagnosis of wave activity over the whole stratosphere and upper troposphere.

Satellite observations of stellar occultations indicate that in the visible band, at the perigee height about 30 km, IGWs and the Kolmogorov turbulence give comparable contributions into the variance of intensity fluctuations (Gurvich and Kan, 2003a, b; Sofieva et al., 2007a, b). In the radio band, due to the larger Fresnel scale, the role of large-scale inhomogeneities with a
steeper 3D spectrum increases. Such inhomogeneities are attributed to IGWs (Kan et al., 2002). This follows from (15) and (16): the decay of variance with increasing wavelength $\sigma_\chi^2 \propto \lambda^{\mu/2-3}$, is stronger for turbulence, $\lambda^{-7/6}$, than for IGWs, $\lambda^{-1/2}$. The relative contribution of IGWs into the variance of amplitude fluctuations with respect to that of isotropic turbulence in the radio band, compared to the visible, increases proportionally to $(\lambda_{GPS}/\lambda_{opt})^{2/3} \approx 5 \cdot 10^3$. This difference is also seen in Figure 1, which shows the amplitude RMS at an altitude of 30 km, if we recollect that in the visible band, the amplitude
fluctuations due to IGWs are additionally restrained by the inner scale of IGWs that exceeds the Fresnel scale by about an order of magnitude.

The statistical analysis of eikonal fluctuations is aggravated by the fact they are non-stationary, and one of the main problem is the determination of the mean profile. We evaluated the eikonal spectrum using two different mean profiles: 1) the model profile and 2) the profile obtained by the sliding averaging of the eikonal profile over an altitude windows with a half width
of $\Delta h$, with the subsequent detrending the eikonal fluctuations. The use of a mean eikonal obtained by sliding averages with $\Delta h$=5 km $> L_W$ and the model profile resulted in very similar spectra.

For strongly anisotropic inhomogeneities, RO signal fluctuations are determined primarily by the vertical structure of inhomogeneities and, accordingly, by the vertical velocity of the ray immersion for different obliquity angles $\alpha$. The comparison of amplitude records taken as a function of time or as a function of perigee height clearly indicates that for different $\alpha$, the tem-
poral dependencies have different characteristic frequencies, while the altitudinal dependencies have nearly the same periods. In the tropical lower troposphere, below the altitude of 7 km, the type of the vertical dependence of the amplitude abruptly changes: the fluctuation frequencies increase, and their magnitude significantly exceeds that at the same altitudes in middle and polar latitudes (Sokolovskiy, 2001, e.g.). In order to obtain a qualitative estimate of the humidity influence, we additionally analyzed the amplitude spectra in the upper and lower troposphere from the COSMIC data in tropics, May 2011, and in middle
and polar latitudes, January 2011. For each latitude band, we chose 30 occultations, with obliquity angles varying from $45°$

to 89°. In the tropics and upper troposphere, at altitudes from 13 down to 8 km, in the amplitude spectra the dominant role is played by anisotropic IGWs. In the lower troposphere, at altitudes from 6 down 1 km, however, the spectra mostly agree with the Kolmogorov turbulence, although some of the spectra have maxima located at higher frequencies, as compared to what is predicted by the theory. This may be a consequence of strong fluctuations, because the relative amplitude fluctuation RMS in tropics, in this altitude range is close to unity. A similar analysis for altitudes from 6 to 1 km, for middle and polar latitudes in January, where the humidity influence was much smaller, indicates that the amplitude spectra mostly correspond to the IGW model, and the fluctuation RMS was smaller than in the tropics, and was equal to 0.2–0.6. This indicates that in the framework of the thin phase screen and weak fluctuation approximations, for the lower troposphere, it is only possible to infer rough estimates of the atmospheric inhomogeneity parameters. A strict quantitative analysis would require more advanced techniques.

The main result of this study consists in the statement that at altitudes above 4-5 km for middle and polar latitudes, and above 7-8 km in the tropics, the dominant contribution into RO signal fluctuations comes from anisotropic inhomogeneities described by the saturated IGW model. This was demonstrated previously by Steiner et al. (2001), who, for the stratosphere, in the altitude range 15–30 km, showed that the temperature fluctuation spectra obtained from GPS/MET observations, in the vertical scale range 2–5 km are in a satisfactory agreement with the saturated IGW model. Pavelyev et al. (2015) analyzed a series of CHAMP occultation events and showed that layered inhomogeneities, as compared to turbulence, play a dominant role in the RO amplitude fluctuations in the stratosphere, and the diffractive slope of the intensity spectra for these inhomogeneities is close to that predicted by the saturated IGW model. Wang and Alexander (2010) and McDonald (2012), analyzing collocated temperature profiles from COSMIC observations, showed that in the stratosphere, the most large-scale dominant temperature perturbations are of wave nature. Gubenko et al. (2008, 2011) developed a method for the determination of the basic characteristics of dominant IGWs, including their intrinsic frequency and phase velocities from vertical profiles of temperature. The method was validated on high-resolution radiosonde observations of temperature and wind and then applied to the analysis of IGW based on temperature profiles retrieved from RO observations in COSMIC and CHAMP missions.

On the other hand, Steiner et al. (2001) only analyzed filtered temperature profiles with scales exceeding 1.5–2 km. The RO signal spectra, as shown in Figures 2–5, have a significantly higher resolution, and the main limitation is imposed by noise. The principle parameters of IGWs are their structure characteristic $C_W^2$ and outer scale $\mathrm{K}_W^{-1}$. Our estimates indicate that humidity fluctuations in middle and polar latitudes are significant below altitudes of 5–6 km; for high altitudes, temperature fluctuations dominate. The relation between $C_{W,dry}^2$ with the traditional IGW parameters is given by (3). The outer scale is introduced in our model in such way that the inhomogeneity spectrum is saturated to a constant for $\kappa_z < \mathrm{K}_W$ (Smith et al., 1987). The temperature variance in the IGW model can be inferred from (11) (Sofieva et al., 2009):

$$\sigma_{\delta T/T}^2 = \frac{4\pi}{3} C_{W,dry}^2 \mathrm{K}_W^{-2} \tag{25}$$

which, in turn, determines the specific potential energy of waves:

$$E_p = \frac{1}{2}\left(\frac{g}{\omega_{B.V.}}\right)^2 \sigma_{\delta T/T}^2 \tag{26}$$

Along with a wide spectrum of saturated IGWs, separate quasi-monochromatic perturbations are detected from spikes in stellar scintillation spectra (Gurvich and Chunchuzov, 2005; Sofieva et al., 2007a). They are, however, rarely observed and do not influence the estimates of statistical moments.

Tsuda et al. (2000); de la Torre et al. (2006); Khaykin et al. (2015) (further references can be found in these papers) studied the global morphology of $E_p$ in the stratosphere using $\sigma^2_{\delta T/T}$ evaluated from temperature profiles retrieved from GPS/MET data. The wave activity can be monitored directly from measurements of amplitude and phase fluctuations of RO signals, using the simple relationships that link them to IGW parameters. A simultaneous determination of structure characteristic and outer scale from RO signal fluctuations allow a more detailed study of IGWs. Adjusting the method of the IGW parameter retrieval from stellar occultations (Gurvich and Kan, 2003a; Sofieva et al., 2007a, 2009), it is possible to derive the structure characteristic and outer scale from amplitude spectra. These parameters can also be inferred from eikonal spectra. Still, it is preferable to use amplitude spectra, which are much more sensitive to refractivity fluctuations: phase variations are proportional to refractivity variations, while amplitude variations are proportional to their second derivative (Rytov et al., 1989b). In addition, strong regular variations of the eikonal with the altitude may introduce significant uncertainties in the lower-frequency region of the eikonal spectrum. On the other hand, for quick estimates, it possible to use variances only. The amplitude variance permits the determination of the structure characteristic (15); the eikonal variance, together with the estimate of the structure characteristic allow the estimate of the outer scale (18). The maximum frequency of amplitude spectra may indicate what inhomogeneity type is essential for the RO signal fluctuations.

## 6   Conclusions

In this study, we presented simple relationships and theoretical estimates of the amplitude and phase variances of RO signal for typical parameters of 3D spectra based on two models: 1) the Kolmogorov turbulence and 2) saturated IGWs. For GPS/MET observation in the altitude range of 4–25 km for middle and polar latitudes, we derived the amplitude and phase fluctuation spectra. Both theoretical and experimental results indicate a dominant role of saturated IGWs in forming the variances and spectra of amplitude and phase fluctuation of RO signal in the stratosphere and upper troposphere, at altitudes above 4–5 km in middle and polar latitudes, and above 7–8 km in the tropics. Simple relationships that link IGW parameters and RO signal fluctuations may serve as a basis for the global monitoring of IGW parameters and activity from RO amplitude and phase observations in the stratosphere and upper troposphere.

*Code availability.*   The code used in this study does not belong to the public domain and cannot be distributed.

*Data availability.*   GPS/MET radio occultation data are freely available. To get access to them, it is necessary to sign up at the website of the CDAAC: http://cdaac-www.cosmic.ucar.edu/cdaac/ (follow the "Sign up" link for further details).

*Competing interests.* The authors declare that they have no conflicts of interest.

*Acknowledgements.* The work of V. Kan and M. E. Gorbunov was supported by Russian Foundation for Basic Research, grant 16-05-00358. We are very grateful to the two reviewers for their appropriate and constructive suggestions.

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
