# Peer review of "Fluctuations of radio occultation signals in sounding the Earth's atmosphere"

_Atmospheric Measurement Techniques, 2017_

## Referee Comment (RC1) · Anonymous Referee #1 · 28 Jul 2017

Review

Title: Fluctuations of radio occultation signals in sounding the Earth's atmosphere
Authors: Valery Kan, Michael E. Gorbunov, Viktoria F. Sofieva

The authors investigate the dominant cause of fluctuation (scintillation) in Earth radio occultation data: clear air turbulence or internal gravity waves. Toward this end they assume empirical spectral forms for the turbulence (Kolmogorov) and for saturated (breaking) internal gravity waves (the "universal" spectrum). They apply a well known analytical method (Rytov approximation for weak scintillation) for relating the power spectra and cross spectrum of log-amplitude and phase scintillations as observed by a radio occultation receiver to the candidate power spectra of refractivity variations. The fundamental key to their test is that clear air turbulence is expected to be isotropic in space whereas gravity waves are anisotropic. If the former is true, the power spectra should depend only transverse motion at ray perigee; if the latter is true, the power spectra should depend solely on ray descent speed. Radio occultation data shows the latter to be true through most of the upper troposphere. The analysis approach of this work is based on a long tradition of radio scintillation analysis begun by Tatarskii, who developed a theory of wave propagation through random media, which is sound. This article should be published pending minor revision, which involves a modicum of additional analysis and justification of approach and significant improvements to grammar.

The authors should present a justification of the spectrum of internal gravity waves (IGW) that they incorporate. For Kolmogorov turbulence, description of its fluctuations by empirical power law is satisfactory. For gravity waves, the case for its description is more difficult. The "universal spectrum" of gravity waves is valid where gravity waves break, due to either Kelvin-Helmholtz or simple convective instability. Its power density spectrum follows a -3 power law in vertical wavenumber; however, its power density spectrum most assuredly does not follow a -3 power law in horizontal wavenumber as incorporated in this manuscript. The power density spectrum in the horizontal rather follows a form that is characteristic of the original source of the gravity waves. Moreover, IGWs break at different levels depending on the strength of their source: moist convection, orographic, jet stream breakdown. Even though it is probably discussed in the some of the papers they cite, the authors should nevertheless offer some justification for assuming the form of the power spectral density of the IGW in horizontal wavenumber as they did and why specifying IGW breaking parameters as a function of height the way they did.

The log-log plots of power spectra the authors present span only one and a half decades, meaning that there is only the slightest constraint on determination of the power law when significant spread between spectra is present. Such is the case in this manuscript. The authors must distinguish between a -3 power law characteristic of IGWs in the log-log plots and a -5/3 power law characteristic of Kolmogorov turbulence, which can be done easily by including a -5/3 line on the power spectral density plots.

Page 1, line 19: "stimulated"
Page 1, line 20: "Currently, RO sounding…"

Page 2, lines 1-3: "The stability of GPS signals, complemented with its global coverage and high vertical resolution, draws the attention of researchers to the study of inhomogeneities in atmospheric refractivity in addition to the retrieval of mean profiles."
Page 2, line 10: "empirical"
Page 2, line 11: "component described"
Page 2, line 12: "the isotropic component as Kolmogorov turbulence"
Page 2, line 18ff: consider calling it "weak scintillation theory" rather than "weak fluctuation theory" throughout the manuscript.
Page 2, line 21-22: "about 30-35 km where residual ionospheric fluctuations and measurement noise become dominant."
Page, line 23: "In the visible band,…" Throughout the text, call it the "visible" band rather than the "optical" band. "Optics" refers to a kind of signal dynamics that spans most frequency bands, including microwave, infrared, visible, and ultra-violet.
Lines 25-26: "In the radio band, the leading cause of the inhomogeneities is IGWs, whose spectra are characterized by a steep power spectral decrease with increasing wavenumber."
Line 31: "dominate the radio signal…"
Line 32: "The aims of this paper are to clarify the role of the two inhomogeneity types and to evaluate their actual contributions…"
Page 3, line 2: "complicated dynamics of lower-tropospheric…"
Line 3: delete "the basic models and approximations"
Line 4: "screen approximation, the weak fluctuation/scintillation theory, and the approximations entailed. In Section 3 we apply these methods to derive…"
Line 17: "statistical average" should be better defined, most likely as "regional average".
Line 17-18: "Refractivity fluctuations depend…"
Page 4, lines 7-8: "are wavevector parameters corresponding to the outer and inner scales, respectively."
Page 4, line 10ff: The vertical wavenumber spectrum for saturated gravity waves is usually referred to as the "universal spectrum". Be sure to cite the original work: Dewan and Good 1986.
Lines 15-27: The idea of "critical anisotropy" is new to me. To what phenomenon does it refer? Be clearer.
Lines 28-30: I'm not sure what this sentence means.
Page 5, line 3: "A = 0.033"
Page 6, equations 4, 5ff: Be clear about the "minus-plus" notation and why you use it. It took me a while to figure out.
Page 7, equation 7: When the thin screen approximation is itself in the small screen approximation with respect to the Earth's curvature, I wouldn't expect there to be any dependence on the Earth's curvature in any equation. So why does the Earth's radius occur in equation 7? Also, write out $\bar{\Psi}$ explicitly.
Equation 8: Does this math also consider distortion of the Fresnel zones by the differential ray bending by the atmosphere's vertical structure?
Page 8ff: Be sure to define precisely the angles $\alpha$, "occultation angle", "obliquity angle". I cannot tell what these angles are.
Page 8, line 14: "or grazing occultation"
Page 11, line 4: "Numerous radiosonde profiles and…"

Page 11, line 8: The value given for $L_W$ is in fact highly variable throughout the global atmosphere. It should have been mentioned somewhere in the introduction that the intention is to qualify RO scintillations as due to turbulence or gravity waves in a gross, global sense.
Page 12, line 3-5: I do not understand this sentence.
Page 13, lines 1-2: "The variances of RO log-amplitude and phase fluctuations…do not contain direct information…" Why can't turbulence be anisotropic at its outer scales? Most of the atmosphere is stably stratified, resisting vertical motion, which means that turbulence would natural seek to extend in the horizontal rather than in the vertical.
Line 4-5: delete "to which the anisotropic…"
Line 5: "This information can be extracted from an ensemble of 1D spectra of RO signal fluctuations, when categorized according to frequency or to vertical wavenumber."
Line 8: What is the oblique movement velocity? Define. "they" should be "the"
Lines 9-10: "for a highly oblique occultation." Delete "due to the geometrical difference…" to the end of the sentence.
Line 16: What is the inclination angle?
Line 22: Linear trends in what? "Figures 2 and 3"
Line 33: "spectral window with variable width"
Line 33-34: Be clear about f. No need to write "Q-factor", a term more appropriate to prescriptions of oscillatory systems.
Page 14, line 1: "Figures 2 and 3"
Line 5ff: Be clear about what you mean when you write "isotropy hypothesis", "anisotropy hypothesis". I believe that the isotropy hypothesis is that the scintillations are caused by Kolmogorov turbulence and that the anisotropy hypothesis is that they are caused by breaking internal gravity waves. The text must be clear on this.
Lines 8-9: "frequency. With increasing occultation angle (???), the maxima systematically…"
Line 10: "all the spectra are peaked near wavenumber 1, which represents the first Fresnel zone…"
Lines 17-19: I don't understand this sentence.
Page 15, line 3: I suspect the "deep oscillations" are a reference to diffraction fringes.
Lines 3-4: "The slope of the spectrum at high frequencies agrees…"
Lines 6-7: This sentence needs clarification. What is \alpha, and what does it have to do with anisotropy?
Line 10: "they mostly exceed the theoretical..."
Line 11: "RMS values prove the validity…"
Line 14: The definition of "eikonal" should be moved much earlier in the document. Either that, or use term "phase" instead throughout the paper. It is a term much more commonly used in the RO community.
Lines 19-20: What are the "first approximation" and the "first term"?
Page 16, line 8: What is a Hann window? Give a reference.
Line 9: "Figures 4 and 5…"
Line 14: "These spectra are in fair agreement…"
Page 17, lines 2-3: "1) isotropic Kolmogorov turbulence, and 2) anisotropic saturated IGWs."
Line 4: "phase with empirical 3D…"
Page 18, line 5: What are "small altitudes"? The boundary layer?
Lines 8-9: "permit a diagnosis of wave activity…"
Line 18: "IGWs are additionally restrained…"

Line 23: Replace "close" with "similar".
Lines 23-24: Remove the sentence. It is obvious.
Line 33: What are "occultation angles"?
Page 19, line 7: Estimates of what?
Line 8: Begin the sentence with "In the stratosphere and upper troposphere, …"
Line 14: "perturbations are sinusoidal."
Line 16: What is "higher resolution"? Higher than what?
Page 20, line 2: "On the other hand, for quick estimates, …. The amplitude variance permits the …"

---

## Referee Comment (RC2) · Anonymous Referee #2 · 24 Aug 2017

Review Fluctuations of radio occultation signals in sounding the Earth's atmosphere

By Valery Kan, Michael E. Gorbunov, and Viktoria F. Sofieva

In the manuscript the previuosly developed for analysis of the stellar occultation data theory and model of turbulent atmopsheric inhomogeneities are applied (after some modernization) to process the GPS/MET RO experimental results. According to the model, the random structure of the atmosphere is represented as an aggregate of two components: (i) the Kolmogorov isotropic; (ii) significantly anisotropic disk-shaped highly flattened horizontal inhomogeneities. The Rytov theory of weak perturbations and the phase-screen method are used to relate the characteristics of IGWs and variations in the amplitude and phase of RO signals propagating through the atmosphere. For strongly anisotropic inhomogeneities, RO signal fluctuations are determined by the vertical velocity of the ray immersion for different occultation angles. High intensity of the anisotropic component in oblique occultations indicated areas of significant activity of IGW's in the atmosphere.
The authors argued that their method can become the basis for global remote sensing of IGW's activity.
The manuscript's material contains valuable information for the audience of the AMT journal.

The publication is possible after significant revision.

**Remarks and suggestions.**

The difficulty of the developed stellar technique transformation for use in radio occultation remote sensing consists in a substantial difference, by several orders of magnitude, of the carrier frequencies, recording and processing methods, and also in the applicable altitude domains in the atmosphere.

1. Radioholograms containing the dependence on time of the amplitude and phase path excess (eikonal) are registered during RO experiments. The scintillations have been measured by GOMOS fast photometers (FP) on board the Envisat satellite at two wavelength $\lambda_B = 499\ nm$; $\lambda_R = 672\ nm$.
So, the RO method has very important additional informative highly accurate phase channel. This channel can be used for identification and separation of the regular layers and turbulence by joint analysis of the RO amplitude and phase data at a single frequency (Pavelyev et al., 2015). The manuscript does not indicate in the reference list or in the text any valuable information on the topic. It is not clear, how one can use the phase RO channel for IGW's analysis.
The suggested in the manuscript technique in the current state does use only the two component statistical model and it is not clear how it separates the possible influence of regular layers from the turbulence contribution in the RO signal. It is well known, that for statistical analysis it is necessary to exclude any systematic influence of regular component on the results.

2. In the manuscript the regular altitude dependence of the refractivity in the atmosphere is described by an exponential model. This is a good approximation for altitudes greater than 20-30 km. However there are clearly defined layers in the stratosphere and troposphere below 30 km. The influence of the regular layers should be taken into account in the formula for the average eikonal estimation (Page 7, line 1, psi=). This is underestimated value. For the troposhere and lower stratosphere this formula should include the bending angle according to the accurate phase path excess formula given by Pavelyev et al., 2015. This concerns also the formula (24) for the refractive attenuation.

3. Besides the above mentioned remarks the paper should contain a clear Figure indicating the main geometrical parameters used in the manuscript (the incidence angle, refractive angle, impact parameter …).

Reference

A.G.Pavelyev, Y.A.Liou, S.S.Matyugov, A.A.Pavelyev, V.N.Gubenko, K.Zhang, and Y.Kuleshov Application of the locality principle to radio occultation studies of the Earth's atmosphere and ionosphere. Atmos. Meas. Tech., 8, 2885–2899, 2015.    www.atmos-meas-tech.net/8/2885/2015/. doi:10.5194/amt-8-2885-2015.

---

## Author Comment (AC1) · 25 Sep 2017

**Response to the reviewers comments on the paper "Fluctuations of radio occultation signals in sounding the Earth's atmosphere" by V. Kan, M. E. Gorbunov, and V. F. Sofieva**

**Reviewer #1**

*The authors should present a justification of the spectrum of internal gravity waves (IGW) that they incorporate. For Kolmogorov turbulence, description of its fluctuations by empirical power law is satisfactory. For gravity waves, the case for its description is more difficult. The "universal spectrum" of gravity waves is valid where gravity waves break, due to either Kelvin-Helmholtz or simple convective instability. Its power density spectrum follows a −3 power law in vertical wavenumber; however, its power density spectrum most assuredly does not follow a −3 power law in horizontal wavenumber as incorporated in this manuscript. The power density spectrum in the horizontal rather follows a form that is characteristic of the original source of the gravity waves. Moreover, IGWs break at different levels depending on the strength of their source: moist convection, orographic, jet stream breakdown. Even though it is probably discussed in the some of the papers they cite, the authors should nevertheless offer some justification for assuming the form of the power spectral density of the IGW in horizontal wavenumber as they did ....*

In the Discussion, we added the following text:

For anisotropic inhomogeneities, we employ an empirical model of saturated IGWs (2). Models of this type are widely used for the analysis of stellar and radio scintillations, the angular dependence of the back-scattering of radar signals, the retrieval of model parameters from occultations etc. 1D vertical and horizontal spectra of this model follow the −3 power. However, air-borne observations (e.g., Nastrom and Gage, 1985; Bacmeister et al., 1996) indicate that the horizontal spectra of temperature fluctuations in the troposphere and stratosphere have a power spectrum with a slope close to −5/3 in a wide range of scales from several km to several hundred km (see also the "saturated-cascade" model of Dewan, 1994). In addition, the model (2) has a constant anisotropy. As noticed in section 2.1, observations of stellar occultations with tangential geometry (Kan et. al, 2014), together with the data about the anisotropy of dominant IGWs (e.g. the description of CRISTA experiment in Ern, et al., 2004; GPS occultations in Wang and Alexander, 2010), have revealed that the anisotropy coefficient is not uniform. It increases from about 10–20 for the IGW breaking scale (10–20 m in the vertical direction) to the saturation value of several hundred for dominant IGWs.

The use of the simple model (2) for the problem in question is justified as follows. As shown above, the most important scales for the IGW model (the Fresnel scale and the outer scale), which determine the RO signal fluctuations, equal or exceed the value of about 1 km in the vertical direction. For inhomogeneities with such vertical scales, the anisotropy significantly exceeds the critical value. Therefore, the amplitude and phase fluctuations do not any longer depend on the anisotropy values and reach the saturation level, as if the inhomogeneities were spherically symmetric. This explains why it is possible to use the model with a strong constant anisotropy. Due to this, the RO observation geometry can be assumed effectively vertical, and the amplitude and phase fluctuations depend only on the vertical structure of saturated IGWs (Eqs. (10) and (14)), which is adequately described by model (2). In some cases, for strongly oblique occultation events, the condition of effectively vertical observation geometry may be broken in the lowest few kilometers due to the strong refraction, which decreases the vertical component of the ray immersion velocity.

Following the ideas of Dalaudier and Gurvich (1997), Gurvich and Chunchuzov (2008) developed an empirical 3D model of saturated IGWs with the anisotropy increasing as a function of the vertical scale. The vertical spectrum follows the −3 power law, while the horizontal

spectrum can have the −5/3 power law for the corresponding choice of anisotropy parameters. This model is in a good agreement with the known air-borne observations of horizontal spectra of IGWs. Scintillation spectra evaluated on the basis of the variable anisotropy model (Gurvich and Chunchuzov, 2008) are in a good agreement with those evaluated on the basis of the constant anisotropy model (2) for effectively vertical occultations (Kan, 2016).

*...and why specifying IGW breaking parameters as a function of height the way they did*
To answer this question, we expanded the last paragraph in page 12, lines 3–7 as follows:
It is known that local profiles of atmospheric inhomogeneities exhibit large natural variability. Furthermore, even their average profiles significantly vary depending on  latitude, season, orography, regions etc. The turbulence structure characteristic for different observations, even in a free atmosphere, may vary by up to two orders of magnitude (e.g., Gracheva and Gurvich, 1980; Wheelon, 2004). A significant variability is observed for the intensity of saturated IGWs (e.g., Sofieva et al., 2007a; Sofieva et al., 2009), which depends both on the sources producing the waves and on the propagation and breaking conditions. Eq. (3) for the saturated IGW only reflects the most general relation between the structure characteristic and the atmospheric stability. The latitudinal variability of the structure characteristic significantly exceeds that of $\omega_{B.V.}^4$ (Sofieva et al., 2009). However, on the average, the variations of refractivity fluctuations and, therefore, the amplitude fluctuations are determined by the exponential decay of the atmospheric density with altitude. Because our work is aimed at a qualitative distinction of the contribution of turbulence and IGWs to the fluctuations of RO signals, we consider only averaged vertical profiles of the structure characteristic of turbulence and IGWs for the theoretical estimates. Quantitative studies of IGW parameters and wave activity for different latitudes, seasons, and regions in the stratosphere and upper troposphere are planned for the future work. Despite possible inaccuracies in the assumed values of the structure characteristic, the variance estimates obtained in this work, definitely indicate the dominant role of saturated IGWs under the conditions in question.

*The log-log plots of power spectra the authors present span only one and a half decades, meaning that there is only the slightest constraint on determination of the power law when significant spread between spectra is present. Such is the case in this manuscript. The authors must distinguish between a -3 power law characteristic of IGWs in the log-log plots and a -5/3 power law characteristic of Kolmogorov turbulence, which can be done easily by including a -5/3 line on the power spectral density plots.*
We updated Figures 2 and 4 with the −5/3 asymptotes, corresponding remarks were added the Figure captions.
In the end of Section "Experimental Fluctuation Spectra of Amplitude and Phase", we added the following paragraph:
The atmospheric inhomogeneity models have not only different anisotropy, but also different slope $-\mu$ of the 3D spectra, which determines the diffractive decay $-\mu + 2$ in the presented spectra of RO amplitudes and phases. The decay is fast, which aggravates the derivation of accurate estimates. Nevertheless, Figures 2-5 indicate that the diffractive decays of the experimental spectra are in a better agreement with the IGW model, as compared to the turbulence model.

*Page 1, line 19: "stimulated"*
*Page 1, line 20: "Currently, RO sounding..."*
*Page 2, lines 1-3: "The stability of GPS signals, complemented with its global coverage and high vertical resolution, draws the attention of researchers to the study of inhomogeneities in atmospheric refractivity in addition to the retrieval of mean profiles."*
*Page 2, line 10: "empirical"*
*Page 2, line 11: "component described"*

*Page 2, line 12: "the isotropic component as Kolmogorov turbulence"*
Corrected.

*Page 2, line 18ff: consider calling it "weak scintillation theory" rather than "weak fluctuation theory" throughout the manuscript.*
Many authors (e.g. Ishimaru, A.: Wave Propagation and Scattering in Random Media. Vol 2; Rytov, S. M., Kravtsov, Y. A., and Tatarskii, V.: Principles of Statistical Radiophysics; Gurvich, A. S. in many works) use terms "weak fluctuations", "smooth perturbations", "Rytov approximation", and "weak scintillations" as equivalent ones. To emphasize the equivalence of "weak fluctuation" and "weak scintillation", we modified the corresponding sentence:
"The upper limit was determined by the radiation shot noise, the lower limit was determined by the applicability condition of the Rytov weak fluctuation/scintillation theory."

*Page 2, line 21-22: "about 30-35 km where residual ionospheric fluctuations and measurement noise become dominant."*
*Page, line 23: "In the visible band,…" Throughout the text, call it the "visible" band rather than the "optical" band. "Optics" refers to a kind of signal dynamics that spans most frequency bands, including microwave, infrared, visible, and ultra-violet.*
Corrected.

*Lines 25-26: "In the radio band, the leading cause of the inhomogeneities is IGWs, whose spectra are characterized by a steep power spectral decrease with increasing wavenumber."*
*Line 31: "dominate the radio signal…"*
*Line 32: "The aims of this paper are to clarify the role of the two inhomogeneity types and to evaluate their actual contributions…"*
*Page 3, line 2: "complicated dynamics of lower-tropospheric…"*
*Line 3: delete "the basic models and approximations"*
*Line 4: "screen approximation, the weak fluctuation/scintillation theory, and the approximations entailed. In Section 3 we apply these methods to derive…"*
Corrected.

*Line 17: "statistical average" should be better defined, most likely as "regional average".*
Yes, it should be the regional and seasonal average estimate.

*Line 17-18: "Refractivity fluctuations depend…"*
This statement refers to the visible band.

*Page 4, lines 7-8: "are wavevector parameters corresponding to the outer and inner scales, respectively."*
Corrected.

*Page 4, line 10ff: The vertical wavenumber spectrum for saturated gravity waves is usually referred to as the "universal spectrum". Be sure to cite the original work: Dewan and Good 1986.*
Corrected.

*Lines 15-27: The idea of "critical anisotropy" is new to me. To what phenomenon does it refer? Be clearer.*
For occultations, the critical value of the anisotropy coefficient $\eta_{cr} = \sqrt{R_e / H_0} \approx 30$ separates moderately anisotropic inhomogeneities with $1 \leq \eta < \eta_{cr}$ and strongly anisotropic inhomogeneities with $\eta > \eta_{cr}$. In the former case, the sphericity of atmospheric layers may not

be taken into account, in the latter case, the sphericity results in the saturation of the eikonal and amplitude fluctuations. Gurvich and Brekhovskikh (2001) introduced this characteristic and the corresponding term. We added here a brief remark: "… the concept of the critical anisotropy will be discussed below (see Eqs. (7) and (8)).

*Lines 28-30: I'm not sure what this sentence means.*

Corrected as: "To obtain the value of the structure characteristic $C_W^2$ in the radio band, $C_{W,dry}^2$ must be multiplied with the coefficient $K^2$, which takes humidity into account (Tatarskii, 1971; …)."

*Page 5, line 3: "A = 0.033"*
Corrected.

*Page 6, equations 4, 5ff: Be clear about the "minus-plus" notation and why you use it. It took me a while to figure out.*
In using this notation, we follow (Rytov et al., 1989). In our opinion, this not only reduces the number of formulas, but also emphasizes the difference between amplitude and phase fluctuations.

*Page 7, equation 7: When the thin screen approximation is itself in the small screen approximation with respect to the Earth's curvature, I wouldn't expect there to be any dependence on the Earth's curvature in any equation. So why does the Earth's radius occur in equation 7? Also, write out $\overline{\Psi}$ explicitly.*
The thin screen introduces the same average phase shift and the same phase fluctuations as the atmosphere along the ray. The phase shift is evaluated by means of the integration of the refractivity along the ray. For inhomogeneities with the anisotropy that exceeds the critical value, the sphericity of the atmosphere must be taken into account, which results in the saturation of fluctuations, because different anisotropic inhomogeneities have different orientation with respect to the line of sight, according to their horizontal position. Therefore, the critical anisotropy is an increasing function of the Earth's radius. Formula (7) gives the expression for the phase (eikonal) fluctuations for the case, when the Earth's sphericity and, therefore, the saturation of fluctuations can be neglected. Formula (8) refers to the case, when the Earth's sphericity must be taken into account. A detailed analysis of the thin screen with account of the Earth's sphericity can be found in (Gurvich, 1984; Gurvich and Brekhovskikh, 2001).
The explicit expression for $\overline{\Psi} = \sqrt{2\pi R_e H_0}\,\overline{N}$ is presented after Eq. (6).

*Equation 8: Does this math also consider distortion of the Fresnel zones by the differential ray bending by the atmosphere's vertical structure?*
The effect of the Fresnel zone compression due to differential regular refraction is approximately taken into account by using the refractive attenuation factor $q$.

*Page 8ff: Be sure to define precisely the angles $\alpha$, "occultation angle", "obliquity angle". I cannot tell what these angles are.*

Now, we uniformly refer to this angle as to the obliquity angle. This angle is defined in the text as follows: "The observation geometry will be determined by the obliquity angle $\alpha$ of the occultation plane, defined as the angle between the immersion direction of the ray perigee and the local vertical in the phase screen."

*Page 8, line 14: "or grazing occultation"*

Corrected.

*Page 11, line 4: "Numerous radiosonde profiles and…"*
Corrected.

*Page 11, line 8: The value given for $L_W$ is in fact highly variable throughout the global atmosphere. It should have been mentioned somewhere in the introduction that the intention is to qualify RO scintillations as due to turbulence or gravity waves in a gross, global sense.*
In the introduction, we added the following remark:
"Our aim is not the quantitative study of RO signal fluctuations, but rather a demonstration of qualitative principal differences between the manifestations of turbulence and IGWs in RO signals."

*Page 12, line 3-5: I do not understand this sentence.*
We extended this paragraph, as specified above.

*Page 13, lines 1-2: "The variances of RO log-amplitude and phase fluctuations…do not contain direct information…" Why can't turbulence be anisotropic at its outer scales? Most of the atmosphere is stably stratified, resisting vertical motion, which means that turbulence would natural seek to extend in the horizontal rather than in the vertical.*
It is true that many researchers complement the Kolmogorov turbulence with anisotropic inhomogeneities at scales approaching the outer scale (e.g., Wheelon, 2004 and further references therein). This allows taking into account the underlying surface in the bottom layer or the influence of the stable stratification in the free atmosphere. We chose the simplest and most commonly used models of 3D inhomogeneities, including the isotropic turbulence, because our aim was not the qualitative retrieval of inhomogeneity parameters, but rather a qualitative estimate of the role of different inhomogeneity types in RO signal fluctuations. Introducing the anisotropy into the largest scales of turbulence will not result in radical changes of the fluctuation estimates: amplitude fluctuations are determined by small-scale inhomogeneities, while the estimates of phase fluctuations are aggravated by the strong regular variations of the phase, as discussed in the paper. Our plan for the future work is to perform quantitative evaluation of the RO signal using 3D models of turbulence and IGWs with variable anisotropy.

*Line 4-5: delete "to which the anisotropic…"*
Corrected.

*Line 5: "This information can be extracted from an ensemble of 1D spectra of RO signal fluctuations, when categorized according to frequency or to vertical wavenumber."*
We updated this sentence as follows: "This information can be extracted from an ensemble of 1D spectra of RO signal fluctuations measured at different obliquity angles, when categorized according to frequency or to vertical wavenumber."

*Line 8: What is the oblique movement velocity? Define.*
We defined it as the velocity of the projection of the ray perigee to phase screen plane.

*"they" should be "the"*
Corrected.

*Lines 9-10: "for a highly oblique occultation." Delete "due to the geometrical difference…" to the end of the sentence.*
Corrected.

*Line 16: What is the inclination angle?*
The obliquity angle.

*Line 22: Linear trends in what? "Figures 2 and 3"*
The mean amplitude profiles were determined by linear fitting.

*Line 33: "spectral window with variable width"*
Corrected.

*Line 33-34: Be clear about $f$. No need to write "Q-factor", a term more appropriate to prescriptions of oscillatory systems.*
We added notation $f$. Instead of Q-factor, we use the term "quality".

*Page 14, line 1: "Figures 2 and 3"*
Corrected.

*Line 5ff: Be clear about what you mean when you write "isotropy hypothesis", "anisotropy hypothesis". I believe that the isotropy hypothesis is that the scintillations are caused by Kolmogorov turbulence and that the anisotropy hypothesis is that they are caused by breaking internal gravity waves. The text must be clear on this.*
Yes, the isotropy hypothesis refers to Kolmogorov turbulence, while the anisotropy hypothesis refers to saturated IGWs. This is clarified in the Figure captions.

*Lines 8-9: "frequency. With increasing occultation angle (???), the maxima systematically..."*
Occultation angle was replaced by obliquity angle throughout the text.

*Line 10: "all the spectra are peaked near wavenumber 1, which represents the first Fresnel zone..."*
Corrected.

*Lines 17-19: I don't understand this sentence.*
We corrected the sentence as follows:
"The variance of amplitude fluctuations weakly depends on the outer scale $L_W$, if it significantly exceeds the Fresnel scale. Nevertheless, the influence of $L_W$ results in a faster than +1 decrease of the spectrum at low frequencies."

*Page 15, line 3: I suspect the "deep oscillations" are a reference to diffraction fringes.*
Yes.

*Lines 3-4: "The slope of the spectrum at high frequencies agrees..."*
Corrected.

*Lines 6-7: This sentence needs clarification. What is $\alpha$, and what does it have to do with anisotropy?*
$\alpha$ is the obliquity angle.

*Line 10: "they mostly exceed the theoretical..."*
*Line 11: "RMS values prove the validity..."*
Corrected.

*Line 14: The definition of "eikonal" should be moved much earlier in the document. Either that, or use term "phase" instead throughout the paper. It is a term much more commonly used in the RO community.*

The eikonal is first defined after formula (5), we complemented the definition with the following text:

"The eikonal, or the optical path, characterizes the propagation media, while the phase also depends on wavelength. In the RO terminology, the excess phase (or phase excess) refers to the eikonal of the observed field with the subtraction of the satellite-to-satellite distance. The excess phase, therefore, characterizes the atmospheric effect in the observed eikonal. The excess phase (eikonal) is modeled by the phase screen. Accordingly, in the observation plane we study the fluctuations for both eikonal and phase."

*Page 15. Lines 19-20: What are the "first approximation" and the "first term"?*

The corrected formulation:

In the first-order approximation of the perturbation method, the eikonal variations are determined by the refractive index variations of the neutral atmosphere (Vorob'ev and Krasil'nikova, 1984).

*Page 16, line 8: What is a Hann window? Give a reference.*

Hann, or cosine window is defined in (Bendat and Piersol, 1986, p. 13). The reference is added.

*Line 9: "Figures 4 and 5…"*
*Line 14: "These spectra are in fair agreement…"*
*Page 17, lines 2-3: "1) isotropic Kolmogorov turbulence, and 2) anisotropic saturated IGWs."*
*Line 4: "phase with empirical 3D…"*

Corrected.

*Page 18, line 5: What are "small altitudes"? The boundary layer?*

Small altitudes are altitudes of a few kilometers. As it follows from eq. (15), for the IGW model $\sigma_\chi^2 \propto q^{3/2}$. The refractive attenuation changes from 1 at large altitudes to approximately 0.15 at 4 km, which partly compensates the increase of the amplitude fluctuation due to larger density at lower altitudes.

We update the text as follows:

"This, together with the strong refractive attenuation at small altitudes, according to (15), significantly reduces the amplitude fluctuations and, therefore, the weak fluctuation condition is met for altitudes down to a few kilometers."

*Lines 8-9: "permit a diagnosis of wave activity…"*
*Line 18: "IGWs are additionally restrained…"*
*Line 23: Replace "close" with "similar".*
*Lines 23-24: Remove the sentence. It is obvious.*

Corrected.

*Line 33: What are "occultation angles"?*

Obliquity angles.

*Page 19, line 7: Estimates of what?*

Estimates of the atmospheric inhomogeneity parameters

*Line 8: Begin the sentence with "In the stratosphere and upper troposphere, …"*

These words can be excluded, because the sentence defines the height ranges.

*Line 14: "perturbations are sinusoidal."*

Sinusoidal form of perturbations is not synonym for their wave nature.

*Line 16: What is "higher resolution"? Higher than what?*
The sentence mentions "high-resolution radiosonde observations".

*Page 20, line 2: "On the other hand, for quick estimates, …. The amplitude variance permits the…*
Corrected.

In addition, we corrected some other typos and references.

---

## Author Comment (AC2) · 25 Sep 2017

**Reviewer #2**

*The difficulty of the developed stellar technique transformation for use in radio occultation remote sensing consists in a substantial difference, by several orders of magnitude, of the carrier frequencies, recording and processing methods, and also in the applicable altitude domains in the atmosphere.*
*1. Radioholograms containing the dependence on time of the amplitude and phase path excess (eikonal) are registered during RO experiments. The scintillations have been measured by GOMOS fast photometers (FP) on board the Envisat satellite at two wavelength $\lambda_B = 499$ nm;*

*$\lambda_R = 672$ nm.*

*So, the RO method has very important additional informative highly accurate phase channel. This channel can be used for identification and separation of the regular layers and turbulence by joint analysis of the RO amplitude and phase data at a single frequency (Pavelyev et al., 2015). The manuscript does not indicate in the reference list or in the text any valuable information on the topic. It is not clear, how one can use the phase RO channel for IGW's analysis.*
*The suggested in the manuscript technique in the current state does use only the two component statistical model and it is not clear how it separates the possible influence of regular layers from the turbulence contribution in the RO signal. It is well known, that for statistical analysis it is necessary to exclude any systematic influence of regular component on the results.*

Yes, unlike optical observations, radio occultations provide not only amplitude, but also phase, i.e. that complex wave field. This opens a prospective for the development and application of advanced radio holographic methods that permit enhancing the accuracy and vertical resolution of refractivity profiles retrieved (e.g. Gorbunov et al., 1998, 2004), as well as obtaining new information about the structure of the atmosphere (Pavelyev et al., 2012, 2015 and further references therein). For example, Pavelyev et al. (2015) demonstrated the power of the locality principle for the localization and estimation of parameters of layered structures and the separation of the contributions of turbulence and layered structures in RO signals.

In our work, we complement the observed amplitude and phase with the observation geometry. Although we use the amplitude and phase separately, the statistical analysis considering the obliquity angle allows us to separate and estimate the contributions of saturated IGWs and turbulence in RO signals. The application of advanced radio holographic methods, in particular the technique of Fourier Integral Operators (Gorbunov and Lauritsen, 2004), for the improvement of accuracy and resolution is our plan for the future work.

We interpret the layered structures discussed by Pavelyev et al. (2015) not as regular or deterministic ones, but as random strongly anisotropic inhomogeneities, approaching spherical layers. They are understood as realizations of a random ensemble of saturated IGWs (e.g. Dewan and Good, 1986; Smith et al, 1987; Gurvich and Brekhovskikh, 2001).

For the wave propagation study, we specify their statistical properties as a 3D model of the spatial spectrum, which is mapped to 2D and 1D spectra of the eikonal fluctuations in the phase screen plane, and, finally, to the fluctuation spectra of the observed amplitude and phase. We separate turbulence and layered structures, using the fact that the amplitude and phase fluctuation spectra of RO signals depend on the anisotropy and the slope of the spatial spectra of the inhomogeneities.

Note, Pavelyev et al. (2015), along with the deterministic description, also applied the statistical approach for the study of realizations of coherent and incoherent components of RO signal, obtained from the combined analysis of the amplitude and phase. For CHAMP occultations event, Pavelyev et al. (2015) showed that layered inhomogeneities play a dominant role in intensity fluctuations in the stratosphere, and that the diffractive slope of the intensity spectra is close to that predicted by the model of saturated IGWs.

Along the line of this remark of the reviewer, we made the following additions.

Page 17, line 14–15, Discussion:

Joint observations of the amplitude and phase of RO signals open new prospective for the development and application of radio holographic methods. These methods allow enhancing the retrieval accuracy and resolution (e.g. Gorbunov et al., 1998; 2004), as well as obtaining new information on the structure of the atmosphere (Pavelyev et al., 2012; 2015 and references therein). In particular, Pavelyev et al. (2015) demonstrated the potential of the locality principle for the localization and estimation of the parameters of layered structures, as well as the separation of the contributions of layered structures and turbulence in RO signals. In our study, we use the power spectra of the observed fluctuations of the amplitude and phase, correlated with the obliquity angle, in order to estimate and separate the contributions of anisotropic inhomogeneities (saturated IGWs) and isotropic turbulence. The application of radio holographic methods for the enhancement of the accuracy and resolution is our plan for future work.

Page 19, line 12–13, after "…with the saturated IGW model":

Pavelyev et al. (2015) analyzed a series of CHAMP occultation events and showed that layered inhomogeneities, as compared to turbulence, play a dominant role in the RO amplitude fluctuations in the stratosphere, and the diffractive slope of the intensity spectra for these inhomogeneities is close to that predicted by the saturated IGW model.

*2. In the manuscript the regular altitude dependence of the refractivity in the atmosphere is described by an exponential model. This is a good approximation for altitudes greater than 20-30 km. However, there are clearly defined layers in the stratosphere and troposphere below 30 km. The influence of the regular layers should be taken into account in the formula for the average eikonal estimation (Page 7, line 1, psi=). This is underestimated value. For the troposphere and lower stratosphere this formula should include the bending angle according to the accurate phase path excess formula given by Pavelyev et al., 2015. This concerns also the formula (24) for the refractive attenuation.*

The use of an enhanced model of the regular atmosphere is critical for the joint use of the amplitude and phase. In our theoretical estimates of the mean eikonal, refraction angle, and refractive attenuation, we employed the simple exponential model of the atmosphere. As shown by Vorob'ev and Krasil'nikova (1994), the relative error for the eikonal and refraction angle, caused by the straight ray approximation, is about 10% for the ray touching the Earth's surface. We were using the simple model, because, as stated in Introduction and Section 3.6, our aim was not the quantitative study of atmospheric inhomogeneities, but rather a demonstration of their qualitative features, in particular, demonstration of the qualitative and principal differences between the manifestations of turbulence and IGWs in RO signals. Strong natural variations of turbulence and saturated IGW parameters significantly exceed all the possible inaccuracies of our approximations.

As already stated above, we adopted the interpretation of the layered structures discussed by Pavelev et al. (2015) as random, strongly anisotropic inhomogeneities described by the saturated IGW model, rather than regular deterministic layers. In the first approximation of the weak fluctuation method, random inhomogeneities do not influence the mean amplitude, phase, and refraction angle (Tatarskii, 1971; Rytov et al., 1989b). The contribution of these inhomogeneities is taken into account by the 1D and 2D eikonal fluctuation spectra.

*3. Besides the above mentioned remarks the paper should contain a clear Figure indicating the main geometrical parameters used in the manuscript (the incidence angle, refractive angle, impact parameter ...).*

*Reference*

*A.G.Pavelyev, Y.A.Liou, S.S.Matyugov, A.A.Pavelyev, V.N.Gubenko, K.Zhang, and Y.Kuleshov Application of the locality principle to radio occultation studies of the Earth's atmosphere and ionosphere. Atmos. Meas. Tech., 8, 2885-2899, 2015. www.atmos-meas-tech.net/8/2885/2015/. doi:10.5194/amt-8-2885-2015.*

Along the line of this remark, we made the following modifications and additions:

Page 5, line 15, after "…to the incident rays":

The occultation geometry has been discussed in many papers: (Vorob'ev and Krasil'nikova, 1994; Ware et al., 1996, Gorbunov and Lauritsen, 2004; Cornman et al., 2004; Pavelyev et al., 2012, and references therein). The phase screen has been discussed in (Hubbard et al., 1978; Gurvich, 1984; Woo et al., 1980: Gurvich and Brekhovskikh, 2001). So we decided not to repeat the Figures from these papers.

Page 7, line 2, after "which is essential, if $\eta \geq \eta_{cr}$.":

Figure 1 in (Gurvich and Brekhovskikh, 2001) provides a good illustration of the influence of the Earth's sphericity upon the eikonal fluctuations in sounding isotropic and anisotropic atmospheric inhomogeneities.

Additional references:

Gorbunov, M. E. and Gurvich, A. S.: Algorithms of inversion of Microlab-1 satellite data including effects of multipath propagation. Int. J. Remote Sensing, 19(12), 2283-2300, 1998.

Pavelyev, A. G., Liou, Y. A., Matyugov, S. S., Pavelyev, A. A., Gubenko, V. N., Zhang, K., and Kuleshov Y.: Application of the locality principle to radio occultation studies of the Earth's atmosphere and ionosphere. Atmos. Meas. Tech., 8, 2885–2899, doi:10.5194/amt-8-2885-2015, 2015.

Pavelyev, A. G., Liou, Y. A., Zhang, K., Wang, C. S., Wickert, J., Schmidt, T., Gubenko, V. N., Pavelyev, A. A., and Kuleshov, Y.: Identification and localization of layers in the ionosphere using the eikonal and amplitude of radio occultation signals, Atmos. Meas. Tech., 5, 1–16, doi:10.5194/amt-5-1-2012, 2012.

---

## Referee Report (RR1)

**Review Fluctuations of radio occultation signals in sounding the Earth's atmosphere**

By Valery Kan, Michael E. Gorbunov, and Viktoria F. Sofieva

General impression

The Author applied a statistical model and theory derived for analysis of the stellar occultation data to analysis of the GPS/MET radio occultation (RO) measurements. The statistical model present a medium as consisting of anisotropic irregularities – a set of pancake-like disks of small thickness, but of large diameter, immersed in a spherically symmetric atmosphere of the Earth. The Authors believe that these inhomogeneities arise from the effect of internal gravity waves (IGW) that are important for the transfer of the kinetic momentum and energy in the atmosphere. In addition, the medium contains isotropic spherical inhomogeneities, their spatial spectrum is described by Kolmogorov's formula. The Authors by use of Rytov approximation and phase screen method connected statistical characteristics of the eikonal and amplitude of the radio occultation (RO) signal variations with spatial spectra of the Kolmogorov's and IGW turbulence. The Authors found that theoretical and experimental results indicate a dominant role of saturated IGWs in forming the variances and spectra of amplitude and phase fluctuation of RO signal in the stratosphere and upper troposphere, at altitudes above 4–5 km in middle and polar latitudes, and above 7–8 km in the tropics.

The Authors stated that IGW parameters and RO signal fluctuations may serve as a basis for the global monitoring of IGW parameters and activity from RO amplitude and phase observations in the stratosphere and upper troposphere.

The Figures support conclusions made by the Authors.

Therefore the article is of significant interest to the audience of the AMT journal.

Some shortcoming:

1. It is known that IGWs have horizontal anisotropy, their fronts are extended along horizontal lines. Therefore, the RO method as compared with stellar occultation has a selectivity that depends on the orientation of the radio beam relative to the IGW wave front.

2. Significant variations of the RO amplitude can be associated not only with multiple propagation, but also with the influence of monochromatic IGWs not described by statistical theory.

3. The discussion on page 4 about the critical anisotropy is indistinct and requires a modification.

Some tautological errors

Page 4 line 5 the anisotropy coefficient *characterized* the ratio of the *characteristic* horizontal and vertical scales,

Page 6 … the measured *variances* and 1D *spectra* of RO signal *fluctuations* with 3D *spectra* of atmospheric refractivity *fluctuation*s for IGW and turbulence *model*s, as well as the *model* profiles of *variances* of 5 RO signal *fluctuations*.

The paper may be published after minor revisions.

---

## Referee Report (RR2)

**Review**

Title: Fluctuations of radio occultation signals in sounding the Earth's atmosphere

Authors: V. Kan, M.E. Gorbunov, V.F. Sofieva

The authors have done a good job in addressing the concerns I raised in my first review. I have two more significant concerns along with several grammatical corrections I missed in the first pass.

My first concern is that I believe the authors misapply the concept of the outer scale in the prescriptions of the internal gravity wave (IGW) and isotropic turbulence power spectral densities. This concern may not actually originate with the authors but with a fairly long list of authors that preceded them in the field of random fluctuations in nonhomogeneous media. Specifically, the outer scale $L$ approximates the correlation length scale of the random atmospheric fluctuations, and it is *not* the vertical depth of the layer in which the fluctuations occur. The trouble in this paper enters in the prescriptions for the outer scale of the IGW spectrum: the authors take 4 km as an upper limit on the outer scale, but depth is more reminiscent of the depth of a breaking gravity wave layer than the correlation length of the breaking gravity waves. One can demonstrate this by estimating the temperature fluctuations associated with breaking gravity waves. Following the same notation as in the paper,

$$\sigma_T \approx \frac{\omega_B^2 T}{g} L \, .$$

Using values typical of the stratosphere and $L = 4$ km, the standard deviation of temperature is approximately 40 K, an absurdly large value and far greater than anything ever seen in the stratosphere. More typical numbers are 1–2 K. This will impact the prescription of the IGW spectral density throughout the paper. I do not expect it to affect the central conclusion of the paper, however.

My second concern is that the authors mistakenly prescribe the outer ($L$) and inner scales ($l$) of the Kolmogorov turbulence independently. Based on mixing length theory and the intuition that the turnover time of the largest eddies in the turbulence are the inverse of the Brunt frequency $\omega_B$, the ratio of the outer to the inner scales of turbulence is given by

$$\left(\frac{L}{l}\right) \approx \left(\frac{\varepsilon_K}{\nu_a \omega_B^2}\right)^{3/4} \approx Re^{3/4}$$

with $Re$ the Reynolds number of the turbulence. The quantities $\varepsilon_K$ and $\nu_a$ are the energy dissipation rate of the turbulence and the kinematic molecular viscosity of the atmosphere. (It is likely that the Reynolds number should be normalized by a large number, ~2000, because that is the critical threshold for the onset of turbulence.) This, too, should affect all of the calculations of the spectral density of the turbulence. Again, I do not expect it to affect the central conclusion of the paper.

**Grammar**
Page 2, line 19: "determined by radiation shot noise"
Page 3, line 7: "not a quantitative study"

Line 10: " weak fluctuation theory"

Line 13: "relative contributions to RO signal fluctuations of isotropic"

Line 19: "approximation of  weak fluctuation"

Line 23: Is the "regional and seasonal statistical average" the relevant average? If so, then $v$ will contain massive contributions from planetary waves and synoptic disturbances that are neither gravity waves nor Kolmogorov turbulence.

Page 4, line 1: "…is locally homogeneous embedded in a spherically symmetric background."

Line 7: "…the anisotropy coefficient defined as…"

Line 10: "The function $\phi$ determines…"

Line 16: "…scales of the IGW model…"

Lines 16-17: "We will show that RO signal fluctuations are determined primarily by the Fresnel scale $\rho_F$ and the outer scale $L$."

Line 18: Spell out "kilometers".

Line 21: "…result in saturation of the eikonal…"

Line 28 (equation 3): Shouldn't the structure constant depend on the outer scale as well?

Lines 29-30: "gravitational acceleration"

Everywhere: Instead of "sphericity", call it the "along-track curvature of the Earth". There is no need for the Earth to be a sphere in this geometry when only relying upon cylindrical symmetry in the vicinity of the occultation.

Page 5, lines 3-4: "the exponent of a purely power-law spectrum must lie between 3 and 5"

Lines 7-8: "refractivity fluctuations and the spectrum (2)…"

Line 15: "several hundred kilometers."

Line 16: " about 3000 km."

Lines 25-26: "regular variation of refraction with altitude. Only when evaluating the phase shift (eikonal) is it necessary to take the Earth's curvature into consideration."

Line 27: "…are considered weak if their variance…"

Line 34: "…especially in the tropics…"

Page 6, lines 4-5: High velocity compared to what? I'm pretty sure the descent velocity is large compared to the atmospheric motions associated with the refractivity inhomogeneities.

Section 3.1: Be a bit more careful in your definitions of y and z.

Page 7, line 7 (equation 6): Isn't the eikonal defined to within an additive constant? If so, how can this equation be correct? Is there a way to write this in terms of the defocusing factor instead?

Page 8, line 5: "…such a steep 3D spectrum ($\mu = 5$) are determined…"

Line 12: "…eikonal fluctuation spectrum…"

Page 9, line 4: "described"

Line 11: "…inner scale in Eq. (2)…"

Equations 15, 16: These look like integral equations but without the integral.

Page 10, line 10: "$A = 0.033$"

Page 11, line 22: "…with a scale height of 2.5 km."

Line 24: As above; the associated temperature fluctuations would be ~40 K. I would guess that the actual outer scale is ~100 m and the breaking gravity waves occur in a 4-km layer.

Line 30: What is "dry optical turbulence"?

Line 31: An outer scale of 1 km for Kolmogorov is associated with an energy dissipation rate integrated in the vertical of 800 W m$^{-2}$. That's an enormous amount, far greater than the energy available to the atmosphere, especially when this is considered as globally representative. I've assumed that the turnover time of the largest eddies is $\omega_B^{-1} \sim$ 50s.

Page 12, line 11: "…and can be attributed to…"

Line 25: "…and region."

Line 27: "Significant variability is observed…"

Page 14, line 16: "The noise spectrum was estimated from…"

Figures 2 and 3 captions: Panels A and B do not *refer* to the isotropy hypothesis and the anisotropy hypothesis. The only difference is the horizontal coordinate ("abscissa"). I believe what you meant to say is something like, "If the refractivity inhomogeneities were isotropic, then the spectral density curves of figures 2 and 3 would be maximized at the same oblique wavenumber (panel A); if instead they are anisotropic, then the curves would be maximized at the same vertical wavenumber (panel B)." It's the strongest argument in the paper.

Page 16, line 26: "…excess phase as the eikonal."

Line 34: "…tenths of a percent…"

Page 17, lines 18-19: The difficulty arises not from the steep slopes but from the lack of range in the spectral interval.

Page 19, line 25: "…open new pathways in the development…"

Page 20, lines 3,4: "degrees", not "degree".

Line 5: "…indicate strong anisotropy…"

Line 10: "…at low altitudes…"

Line 13: "visible occultations."

Line 22: "visible", not "optics"

Line 29: "The use of a mean eikonal obtained by sliding averages with…"

Line 35: "…have nearly the same periods."

Page 21, line 1: "In the tropical lower troposphere…"

Line 13: "…in the framework of the thin phase screen and weak fluctuation approximations…"

Lines 18-19: "This was demonstrated previously by Steiner et al. (2001), who, for the stratosphere, in the altitude range 15–30 km, …"

---

## Author Response (AR2)

Response to Reviewer #1

*My first concern is that I believe the authors misapply the concept of the outer scale in the prescriptions of the internal gravity wave (IGW) and isotropic turbulence power spectral densities. This concern may not actually originate with the authors but with a fairly long list of authors that preceded them in the field of random fluctuations in nonhomogeneous media. Specifically, the outer scale L approximates the correlation length scale of the random atmospheric fluctuations, and it is **not** the vertical depth of the layer in which the fluctuations occur. The trouble in this paper enters in the prescriptions for the outer scale of the IGW spectrum: the authors take 4 km as an upper limit on the outer scale, but depth is more reminiscent of the depth of a breaking gravity wave layer than the correlation length of the breaking gravity waves. One can demonstrate this by estimating the temperature fluctuations associated with breaking gravity waves. Following the same notation as in the paper,*

$$\sigma_T = \frac{\omega_B^2 T}{g} L$$

*Using values typical of the stratosphere and $L = 4$ km, the standard deviation of temperature is approximately 40 K, an absurdly large value and far greater than anything ever seen in the stratosphere. More typical numbers are 1–2 K. This will impact the prescription of the IGW spectral density throughout the paper. I do not expect it to affect the central conclusion of the paper, however.*

We adopted the external scale value $L_W = 4$ km according to (Smith et al., 1987; Tsuda et al., 1991). From Eqs. (2) and (3), one can readily derive:

$$\sigma_T^2 = T^2 \int \Phi_W(\boldsymbol{\kappa}) d^3\boldsymbol{\kappa} = \frac{4\pi T^2 C_W^2}{3K^2} = \frac{T^2 C_W^2 L^2}{3\pi} = \frac{\beta \omega_B^4 T^2 L^2}{4\pi^2 g^2},$$

(cf. Eq. (25)) and, therefore,

$$\sigma_T = \frac{\sqrt{\beta}}{2\pi} \frac{\omega_B^2 T}{g} L,$$

The additional factor of $\sqrt{\beta}/2\pi$ with $\beta = 0.1$ has a value of about 0.05, which makes the temperature variance estimate much more realistic, i.e. just about 2 K, conforming with the values cited by the reviewer. See also the discussion of the next remark.

*My second concern is that the authors mistakenly prescribe the outer ($L$) and inner scales ($l$) of the Kolmogorov turbulence independently. Based on mixing length theory and the intuition that the turnover time of the largest eddies in the turbulence are the inverse of the Brunt frequency $\omega_B$, the ratio of the outer to the inner scales of turbulence is given by*

$$\left(\frac{L}{l}\right) \approx \left(\frac{\varepsilon_K}{\nu_a \omega_B^2}\right)^{3/4} \approx \mathrm{Re}^{3/4}$$

*with $\mathrm{Re}$ the Reynolds number of the turbulence. The quantities $\varepsilon_K$ and $\nu_a$ are the energy dissipation rate of the turbulence and the kinematic molecular viscosity of the atmosphere. (It is likely that the Reynolds number should be normalized by a large number, ~2000, because that is the critical threshold for the onset of turbulence.) This, too, should affect all of the calculations of the spectral density of the turbulence. Again, I do not expect it to affect the central conclusion of the paper.*

The above formula must be understood as a qualitative estimate rather than an exact relation between $L$ and $l$, and this applies to any theoretical relations between inner/outer scale, temperature/velocity gradients etc. The reason is that the structure of turbulence at scales approaching the outer scale is not well

understood (e.g. Tatarskii, 1971; Wheelon, 2004). In the first place, this applies to the free atmosphere (upper troposphere and stratosphere). This explains why we specified the outer scale basing on experimental data and chose the upper boundary of its possible values for the height range of 4–30 km (Wheelon, 2004).

This allowed us to demonstrate that even for the maximum possible value of $K_K^{-1}$, the contribution of isotropic turbulence into RO signal fluctuations is small compared to that of saturated IGWs.

**Grammar**
*Page 2, line 19: "determined by radiation shot noise"*
*Page 3, line 7: "not a quantitative study"*
*Line 10: "the weak fluctuation theory"*
*Line 13: "relative contributions to RO signal fluctuations of isotropic"*
*Line 19: "approximation of the weak fluctuation"*
This was corrected.

*Line 23: Is the "regional and seasonal statistical average" the relevant average? If so, then $\nu$ will contain massive contributions from planetary waves and synoptic disturbances that are neither gravity waves nor Kolmogorov turbulence.*
We assume that the spatial and temporal scale of these perturbations significantly exceed the characteristic scales of RO signal fluctuations, including the Fresnel zone, as well as the outer and inner scales of the inhomogeneities. This allows disregarding the large-scale processes. A corresponding remark was added to the text.

*Page 4, line 1: "…is locally homogeneous embedded in a spherically symmetric background."*
The random field is assumed to be statistically homogeneous on a sphere (Gurvich, 1984).

*Line 7: "…the anisotropy coefficient defined as…"*
*Line 10: "The function $\phi$ determines…"*
*Line 16: "…scales of the IGW model…"*
*Lines 16-17: "We will show that RO signal fluctuations are determined primarily by the Fresnel scale $\rho_F$ and the outer scale $L$."*
*Line 18: Spell out "kilometers".*
*Line 21: "…result in saturation of the eikonal…"*
This was corrected.

*Line 28 (equation 3): Shouldn't the structure constant depend on the outer scale as well?*
In our model of saturated IGWs, the anisotropy coefficient, structure characteristic, and the outer scale are independent parameters. Corresponding remark was added after Eq. (2).

*Lines 29-30: "gravitational acceleration"*
*Everywhere: Instead of "sphericity", call it the "along-track curvature of the Earth". There is no need for the Earth to be a sphere in this geometry when only relying upon cylindrical symmetry in the vicinity of the occultation.*
*Page 5, lines 3-4: "the exponent of a purely power-law spectrum must lie between 3 and 5"*
*Lines 7-8: "refractivity fluctuations and the spectrum (2)."*
*Line 15: "several hundred kilometers."*
*Line 16: " about 3000 km."*
*Lines 25-26: "regular variation of refraction with altitude. Only when evaluating the phase shift (eikonal) is it necessary to take the Earth's curvature into consideration."*
*Line 27: "…are considered weak if their variance…"*

*Line 34: "...especially in the tropics..."*
This was corrected.

*Page 6, lines 4-5: High velocity compared to what? I'm pretty sure the descent velocity is large compared to the atmospheric motions associated with the refractivity inhomogeneities.*
Yes, we added a corresponding clarification.

*Section 3.1: Be a bit more careful in your definitions of $y$ and $z$.*
We added a definition of $y$.

*Page 7, line 7 (equation 6): Isn't the eikonal defined to within an additive constant? If so, how can this equation be correct? Is there a way to write this in terms of the defocusing factor instead?*
It is correct that in RO experiments, eikonal is measured to within an unknown additive constant. However, we define the atmospheric eikonal without this uncertainty, as the integral of the atmospheric refractivity $N$ along the ray. According to this definition, the mean eikonal $\overline{\Psi}$ for the exponential model of the atmosphere equals $\sqrt{2\pi R_e H_0}\,\overline{N}$. When studying experimental eikonal fluctuation spectra, the unknown constant will be included into the definition of the mean smoothed eikonal and will thus vanish.

*Page 8, line 5: "...such a steep 3D spectrum ($\mu = 5$) are determined..."*
*Line 12: "...eikonal fluctuation spectrum..."*
*Page 9, line 4: "described"*
*Line 11: "...inner scale in Eq. (2)...":*
This was corrected.
*Equations 15, 16: These look like integral equations but without the integral.*
Missing integrals are inserted.

*Page 10, line 10: " $A = 0.033$ "*
*Page 11, line 22: "...with a scale height of 2.5 km..."*
This was corrected.

*Line 24: As above, the associated temperature fluctuations would be ~40 K. I would guess that the actual outer scale is ~100 m and the breaking gravity waves occur in a 4-km layer.*
See the above discussion of $\sigma_T$, for which our model gives much smaller, realistic values.

*Line 30: What is "dry optical turbulence"?*
The term "optical turbulence" refers to the turbulence as a distorting factor for the propagation of optical radiation. "Dry optical turbulence" refers to the fact that in the optical domain refractivity only includes the "dry term" $N = CP/T$. The word "dry" can be omitted.

*Line 31: An outer scale of 1 km for Kolmogorov is associated with an energy dissipation rate integrated in the vertical of 800 W m⁻². That's an enormous amount, far greater than the energy available to the atmosphere, especially when this is considered as globally representative. I've assumed that the turnover time of the largest eddies is 50 s.*
As already noticed, the we adopted the outer scale of 1 km as the upper bound of its possible values, in order to demonstrate that even for such a large outer scale, the contribution of the Kolmogorov turbulence to RO signal fluctuations is small compared to that of saturated IGWs. A remark along these lines was added to the text.

*Page 12, line 11: "…and can be attributed to…"*
*Line 25: "…and region…"*
*Line 27: "Significant variability is observed…"*
*Page 14, line 16: "The noise spectrum was estimated from…":*
This was corrected.

*Figures 2 and 3 captions: Panels A and B do not **refer** to the isotropy hypothesis and the anisotropy hypothesis. The only difference is the horizontal coordinate ("abscissa"). I believe what you meant to say is something like, "If the refractivity inhomogeneities were isotropic, then the spectral density curves of figures 2 and 3 would be maximized at the same oblique wavenumber (panel A); if instead they are anisotropic, then the curves would be maximized at the same vertical wavenumber (panel B)." It's the strongest argument in the paper.*
We corrected the figure captions along these lines.

*Page 16, line 26: "…excess phase as the eikonal."*
*Line 34: "…tenths of a percent…"*
This was corrected.

*Page 17, lines 18-19: The difficulty arises not from the steep slopes but from the lack of range in the spectral interval.*
One is linked to the other. The diffractive slope of $-\mu + 2$ in the presented fluctuation spectra must cover the range from the Fresnel zone (about 1 km) to the inner scale (tenths of meters for IGWs). Due the fast decay, the noise limits the spectral range at 2 octaves above the Fresnel wavenumber. We updated the formulation in the text along these lines.

*Page 19, line 25: "…open new pathways in the development…"*
*Page 20, lines 3, 4: "degrees", not "degree".*
*Line 5: "…indicate strong anisotropy…"*
*Line 10: "…at low altitudes…"*
*Line 13: "visible occultations."*
*Line 22: "visible", not "optics"*
*Line 29: "The use of a mean eikonal obtained by sliding averages with…"*
*Line 35: "…have nearly the same periods."*
*Page 21, line 1: "In the tropical lower troposphere…"*
*Line 13: "…in the framework of the thin phase screen and weak fluctuation approximations…"*
*Lines 18-19: "This was demonstrated previously by Steiner et al. (2001), who, for the stratosphere, in the altitude range 15–30 km, …"*
This was corrected.

Reviewer #2

*Some shortcoming:*

*1. It is known that IGWs have horizontal anisotropy; their fronts are extended along horizontal lines. Therefore, the RO method as compared with stellar occultation has a selectivity that depends on the orientation of the radio beam relative to the IGW wave front.*
We do not understand how the second sentence follows from the first one. We do not see any reasons for the RO method to have any specific selectivity depending on the observation geometry orientation with respect to IGW wave front. We use a stochastic description of gravity waves in our paper. Random realizations of internal gravity waves generate asymmetric irregularities. There are many reasons for the asymmetry, including the wind shear, changes of the density gradient with height, influence of Coriolis forces etc. However, all these factors play a minor role while considering the statistical description in the range of scales corresponding to saturated IGW, which, as an ensemble of a large number of events with different observation geometry, have a probability distribution approaching the isotropic one.

*2. Significant variations of the RO amplitude can be associated not only with multiple propagation, but also with the influence of monochromatic IGWs not described by statistical theory.*
Along with a wide spectrum of saturated IGWs, separate quasi-monochromatic perturbations are detected from spikes in stellar scintillation spectra (Gurvich, A. and Chunchuzov, I.: Estimates of characteristic scales in the spectrum of internal waves in the stratosphere obtained from space observations of stellar scintillations, J. Geophys. Res., 110, D03114, doi:10.1029/2004JD005199, 2005; Sofieva et al., 2007a). Because these quasi-monochromatic structures are rarely observed, they do not influence the estimates of statistical moments. A remark along these lines was added to the text.

*3. The discussion on page 4 about the critical anisotropy is indistinct and requires a modification.*
The critical anisotropy is explained by the fact that for inhomogeneities inclined with respect to the line of sight, a ray is only influenced by a limited horizontal piece of each inhomogeneity. This remark was added in page 4.
As stated in page 4, a more detailed discussion of the critical anisotropy can be found in page 7, in the discussion of Eqs. (6)–(8). We also make a reference to Figure 1 in (Gurvich and Brekhovskikh, 2001), which 
[revised manuscript text omitted]

$$
$$
= \frac{1}{2}\left\{ \tilde{B}_S\left(\Delta z, \Delta y\right) \mp \frac{k\gamma}{4\pi x_1 q^{1/2}} \iint \tilde{B}_S\left(\Delta z', \Delta y'\right) \sin\left[\frac{k\gamma}{4x_1 q}\left(\Delta z' - \Delta z\right)^2 + \frac{k\gamma}{4x_1}\left(\Delta y' - \Delta y\right)^2\right] d\Delta z' d\Delta y'\right\}
$$
$$
B_{\chi S}\left(\Delta z_0, \Delta y_0\right) =
$$
$$
= \frac{1}{2}\frac{k\gamma}{4\pi x_1 q^{1/2}} \iint \tilde{B}_S\left(\Delta z', \Delta y'\right) \cos\left[\frac{k\gamma}{4x_1 q}\left(\Delta z' - \Delta z\right)^2 + \frac{k\gamma}{4x_1}\left(\Delta y' - \Delta y\right)^2\right] d\Delta z' d\Delta y' \tag{4}
$$

where $\chi$ is the logarithmic amplitude, $S$ is the phase, $k = 2\pi/\lambda$, axis $x_0$ is collinear with the incident ray direction, axis $y_0$ is transverse, and axis $z_0$ is vertical, $\gamma = \frac{x_t + x_1}{x_t}$, $x_t$ is the distance from the transmitter to the phase screen, $x_1$ is the distance from the phase screen to the receiver, $q$ is refractive attenuation coefficient, $\Delta z, \Delta y$ are the scales in the phase screen, defined as the coordinate differences of the phase stationary points, and linked to the corresponding scales in the observation plane by the following relationships: $\Delta z = \frac{q}{\gamma}\Delta z_0, \Delta y = \frac{1}{\gamma}\Delta y_0$, $\tilde{B}_S\left(\Delta z, \Delta y\right)$ is the correlation function of the phase in the phase screen, $B_{\chi S}$ is the mutual correlation function of the logarithmic amplitude and phase. The negative sign in the upper formula in (4) applies to the amplitude, and the positive sign applies to the phase.

Taking the Fourier transform, we arrive at the following expressions for the 2D fluctuation spectra of the received signal:

$$
F_{\chi,S}\left(\kappa_z, \kappa_y\right) = \frac{k^2}{2}\left\{1 \mp \cos\left[\frac{x_1 q}{k\gamma}\left(\kappa_z^2 + q^{-1}\kappa_y^2\right)\right]\tilde{F}_\varphi\left(\kappa_z, \kappa_y\right)\right\}
$$
$$
F_{\chi S}\left(\kappa_z, \kappa_y\right) = \frac{k^2}{2}\sin\left[\frac{x_1 q}{k\gamma}\left(\kappa_z^2 + q^{-1}\kappa_y^2\right)\right]\tilde{F}_\varphi\left(\kappa_z, \kappa_y\right) \tag{5}
$$

[revised manuscript text omitted]